# A first assessment of the distribution and abundance of large pelagic species at Cocos Ridge seamounts (Eastern Tropical Pacific) using drifting pelagic baited remote cameras

**Marta Cambra** 1,2 *, **Frida Lara-Lizardi** 3,4, **César Peñaherrera-Palma** 3, **Alex Hearn** 3,5, **James T. Ketchum** 3,4,6, **Patricia Zarate** 3,7, **Carlos Chacón** 8, **Jenifer Suárez-Moncada** 9, **Esteban Herrera** 10, **Mario Espinoza** 2,3,11

1 Programa de Posgrado en Biología, Universidad de Costa Rica, San Pedro, San José, Costa Rica, 2 Centro de Investigación en Ciencias del Mar y Limnología, Universidad de Costa Rica, San Pedro, San José, Costa Rica, 3 MigraMar, Sir Francis Drake Boulevard, Olema, California, United States of America, 4 Pelagios Kakunjá, La Paz, Baja California Sur, México, 5 Galapagos Science Center, Universidad San Francisco de Quito, Quito, Ecuador, 6 Centro de Investigaciones Biológicas del Noroeste-CIBNOR, La Paz, Baja California Sur, México, 7 División de Investigación Pesquera, Instituto de Fomento Pesquero, Valparaíso, Chile, 8 Fundación Pacífico, Sabana Norte, San José, Costa Rica, 9 Dirección del Parque Nacional Galápagos, Isla Santa Cruz, Ecuador, 10 Área de Conservación Marina Cocos, Heredia, Costa Rica, 11 Escuela de Biología, Universidad de Costa Rica, San Pedro, San José, Costa Rica

* m.cambra.agusti@gmail.com, marta.cambra@ucr.ac.cr

**Data Availability Statement:** The authors confirm that all data underlying the findings are fully

## Abstract

Understanding the link between seamounts and large pelagic species (LPS) may provide important insights for the conservation of these species in open water ecosystems. The seamounts along the Cocos Ridge in the Eastern Tropical Pacific (ETP) ocean are thought to be ecologically important aggregation sites for LPS when moving between Cocos Island (Costa Rica) and Galapagos Islands (Ecuador). However, to date, research efforts to quantify the abundance and distribution patterns of LPS beyond the borders of these two oceanic Marine Protected Areas (MPAs) have been limited. This study used drifting-pelagic baited remote underwater video stations (BRUVS) to investigate the distribution and relative abundance of LPS at Cocos Ridge seamounts. Our drifting-pelagic BRUVS recorded a total of 21 species including elasmobranchs, small and large teleosts, dolphins and one sea turtle; of which four species are currently threatened. Depth of seamount summit was the most significant driver for LPS richness and abundance which were significantly higher at shallow seamounts (< 400 m) compared to deeper ones (> 400m). Distance to nearest MPA was also a significant predictor for LPS abundance, which increased at increasing distances from the nearest MPA. Our results suggest that the Cocos Ridge seamounts, specifically Paramount and West Cocos which had the highest LPS richness and abundance, are important aggregation sites for LPS in the ETP. However, further research is still needed to demonstrate a positive association between LPS and Cocos Ridge seamounts. Our findings showed that drifting pelagic BRUVS are an effective tool to survey LPS in fully pelagic ecosystems of the ETP. This study represents the first step towards the standardization of this technique throughout the region.

available without restriction. All relevant data are within the paper and its Supporting information files.

**Funding:** This study was financially supported by Fundación Pacífico, a non-profit regional marine conservation fund. The data collection for this project would not have been possible without the generous support of Mr. Ted Waitt who donated the use of the Plan B vessel (including crew, equipment, etc.) to collect the field data and the support of the Waitt Foundation to cover additional costs of the expedition. Additional funding from the Shark Conservation Fund, the Helmsley Charitable Trust, Iris and Michael Smith and The Galapagos Conservation Trust also supported this project. The funders had no role in study design, data collection and analysis, decision to publish, or preparation of the manuscript.

**Competing interests:** The authors have declared that no competing interests exist.

# Introduction

Quantifying the spatial distribution and abundance of pelagic species (hereafter referred to as species that spend the majority of their lives inhabiting the upper layers of the water column in oceanic waters) is critical to effectively manage and protect their populations in the open oceans [1–3]. Overexploitation of the open ocean by industrial fisheries is driving many large pelagic species (reported common length in [4] > 1m, LPS) such as elasmobranchs, teleosts, sea turtles and cetaceans to dangerously low levels [2,5], raising global concerns about the potential top-downs effects on marine ecosystems [6,7]. Seamounts have been recognized as productive and unique features in open water-systems where highly migratory LPS tend to aggregate, thus becoming vulnerable areas to overfishing [8,9]. Understanding the link between seamounts and LPS may be fundamental to identify regional hotspots of biological production, and therefore, to guide management and conservation efforts in open water eco-systems [10].

The Cocos Ridge is a chain of seamounts in the Eastern Tropical Pacific (ETP) ocean that connects Cocos Island (Costa Rica) and the Galapagos archipelago (Ecuador) (Fig 1) [11]. These two oceanic island groups are considered biodiversity hotspots in the ETP because of their high apex predator biomass [12,13]. They are also important no-take Marine Protected Areas (MPAs) and UNESCO World Heritage Sites within the ETP [14,15]. Previous studies have shown a higher degree of movement connectivity between Cocos and Galapagos Islands relative to other regions of the ETP, suggesting that LPS may be using this area as a migratory corridor [16–20]. The Cocos Ridge seamounts are thought to be ecologically important aggregation sites for LPS during their migratory movements between both MPAs [20]. However, most research effort on LPS in the ETP have concentrated inside MPAs [21–24], and there is currently limited information available on the role that seamounts play on the population structure and dynamics of LPS in this region (although see [25]).

As populations of LPS continue to decrease in the ETP [21,23,28], there is a greater need to survey the pelagic ecosystem beyond the borders of MPAs to effectively guide marine spatial planning [20,29]. Information outside the protection boundaries of Cocos and Galapagos Islands is scarce and restricted to fishery dependent data [30–32] or to movement studies on sharks [16–19,25], teleosts [33] and sea turtles [34,35]. Despite the valuable information acquired from biotelemetry to understand individual habitat preferences, movements and migrations, this technique relies on the catch of a high number of individuals from various species in order to understand how the pelagic community is distributed in the open ocean [36]. Furthermore, such studies can be invasive, expensive and logistically challenging [37]. Fisheries data also present some limitations as they are usually biased by temporally and spatially uneven sampling effort, gear selectivity and lack of robust reports [38,39].

Drifting-pelagic baited remote underwater video stations (BRUVS) have demonstrated a promising potential for studying pelagic wildlife in open water ecosystems [40–42]. Drifting-pelagic BRUVS are an adaptation of the benthic BRUVS where an anchoring system is no longer needed, thus enabling dynamic sampling over deep and topographically complex pelagic areas [41]. The odor of the bait triggers bait-search behavior in nearby fish assemblages, increasing the probability of detecting predatory species in the vicinity of the BRUVS [43]. A reduced amount of zeros (i.e. less absences from count data) derived from bait use increases the statistical power of BRUVS compared to traditional survey techniques [44–46]. Although studies using drifting-pelagic BRUVS are scarce [41,42,47], this technique offers a powerful framework to overcome the difficulties associated to effectively survey pelagic assemblages [40]. For example, drifting-pelagic BRUVS units can be simultaneously deployed reducing the survey effort while generating permanent high definition images on species composition,

behavior and relative abundance at different depth levels and for long periods of time [41]. Additionally, they are affordable and easy to operate allowing the participation of non-expert stakeholders into field work. In this study, we used drifting-pelagic BRUVS for the first time in offshore waters of the ETP to investigate the distribution and relative abundance of LPS at Cocos Ridge seamounts. This study may serve as an important baseline reference on the future use of drifting-pelagic BRUVS at a regional level.

## Methodology

### Study area

The Cocos Ridge is an underwater mountain range located in the Northwestern Panama basin of the ETP, which originated more than 30 million years ago as a result of volcanic activity from the Galapagos Ridge hot spot [11]. The Cocos Ridge rises about 2000 m above the seafloor and extends more than 1000 km from the Galapagos Islands to Cocos Island, and from Cocos Island to the Pacific coast of Costa Rica [11,48,49]. Although the depth and total number of seamounts along the Cocos Ridge is not available on global bathymetric databases, there are at least 14 seamounts that have been identified in this region [20,48]. All the seamounts along the Cocos Ridge are located within the Exclusive Economic Zones (EEZs) of either Ecuador or Costa Rica (Fig 1).

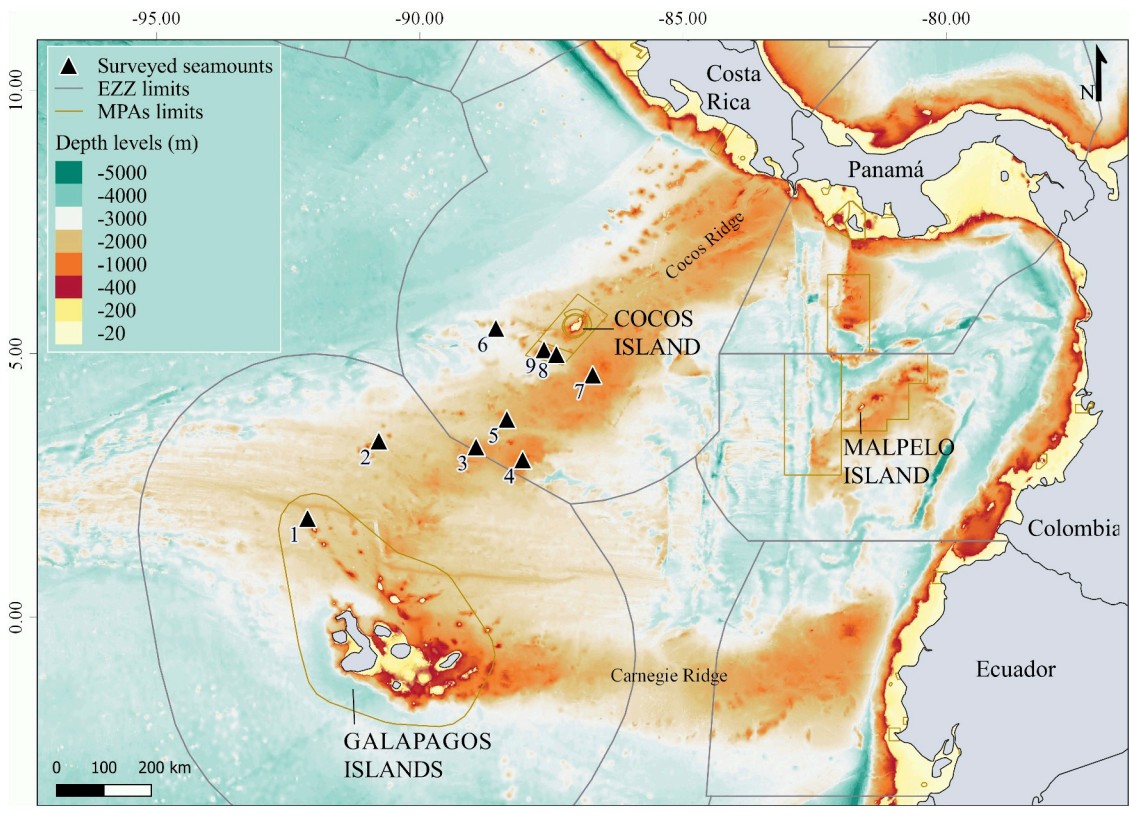

**Fig 1. Location of seamounts surveyed along the Cocos Ridge between Cocos Island and the Galapagos Islands.** Numbers indicate surveyed seamounts: (1) NW Darwin; (2) Paramount; (3) Medina 1; (4) Medina 2; (5) Medina 3; (6) West Cocos; (7) East Cocos; (8) Las Gemelas 1; (9) Las Gemelas 2. Limits of the Marine Protected Areas (MPAs) in the Economic Exclusive Zone (EEZ) of each country are shown. Bathymetry data was obtained from ETOPO1 1 Arc-Minute Global Relief Model [26,27] under a CC BY license, with permission from NOAA National Centers for Environmental Information, original copyright 2009.

The variable and dynamic oceanographic conditions surrounding the waters of Cocos Ridge seamounts are mainly attributed to the southern oscillation of the Intertropical Convergence Zone, which is also associated to seasonal changes on wind patterns and the convergence of different ocean currents [20,50]. There are several major oceanic current systems converging in this region, resulting in a unique variety of tropical and temperate marine life [51]. The North Equatorial Current (east—west), the South Equatorial Current (east—west) and the Equatorial Countercurrent (west—east) are the main surface currents affecting the Cocos Ridge seamounts [52]. Although the central region of the ETP is characterized by warm waters (average 27.5˚C), the Humboldt Current (the southern limit of the ETP) and the Cromwell undercurrent (west—east) can bring cooler waters (down to ~18˚C and ~12˚C, respectively) potentially affecting seamounts closer to Galapagos Islands [52]. Oceanographic variability is accentuated by the increase of the average sea temperature during the El Niño–Southern Oscillation Phenomenon, which occurs at irregular intervals of 2–7 years [53]. Chlorophyll-a concentrations along the marine corridor between Cocos and Galapagos Islands respond to seasonal variations typically oscillating between 0.15 and 0.22 mg m$^{-3}$ [20]. The ETP is also characterized by a shallow thermocline (often at 25 m) above a permanent barrier of cold hypoxic water which may limit the available physical habitat for some predator species [54,55].

## Sampling method

Nine seamounts along the Cocos Ridge were surveyed from April 3$^{rd}$ to April 11$^{th}$ of 2018 (Fig 1) when the ETP was transitioning from La Niña (cooler-than-average-waters) to ENSO-neutral conditions (sea surface temperatures near the long-term average) [56]. The survey was conducted under the research permit PC-24-18 issued by the Galapagos National Park Directorate. Permits to survey seamounts in Costa Rica were not necessary because none of them were located inside Cocos Island National Park. Seamount selection was based on the depth of each seamount summit. We prioritized the shallowest seamounts in the study area to increase the probability of LPS observation rates in subsurface waters (10 and 25 m deep) based on results from [57] and [42], where shallower seamounts (< 400 m) showed significant effects on the aggregation of LPS. The depth of each seamount was previously determined using the Seamount Catalog of EarthRef [58] and the National Center for Environmental Information of the National Oceanic and Atmospheric Administration (NOAA) [59]. The depth sounder of the research vessel used during the expedition provided more accurate locations for seamounts shallower than 600 m (depth sounder maximum capacity). Additional information regarding seamount bathymetry was accessed during the expedition from boat navigation charts and from offline nautical charts [60].

The drifting-pelagic BRUVS used in this study are an adaptation of designs used elsewhere by [61,62]. Our design consists of a triangle shaped stainless-steel frame that supports a single high-definition GoPro Hero 4 camera encased in an underwater housing (Fig 2). Each camera was provided with a backpack battery to extend its recording time between 2 and 3 hours. Cameras were set to record at 60 frames per second/1080p resolution in wide field of view to maximize detection rates. All units had a baited arm to hold a perforated PVC bait container. A total of 1.5 kg of yellowfin tuna (*Thunnus albacares*) was used per each BRUVS during approximately 2.5 hr of soak time (mean ± SD: 140 ± 13.4 min) considering the little signs of species accumulation captured in the open ocean using drifting-pelagic BRUVS after 180 minutes [37,41]. Once thawed, the bait was cut into approximately 5 cm pieces and lightly crushed once inserted into the perforated PVC container before each deployment. Drifting pelagic BRUVS were manually launched and retrieved from the vessel.

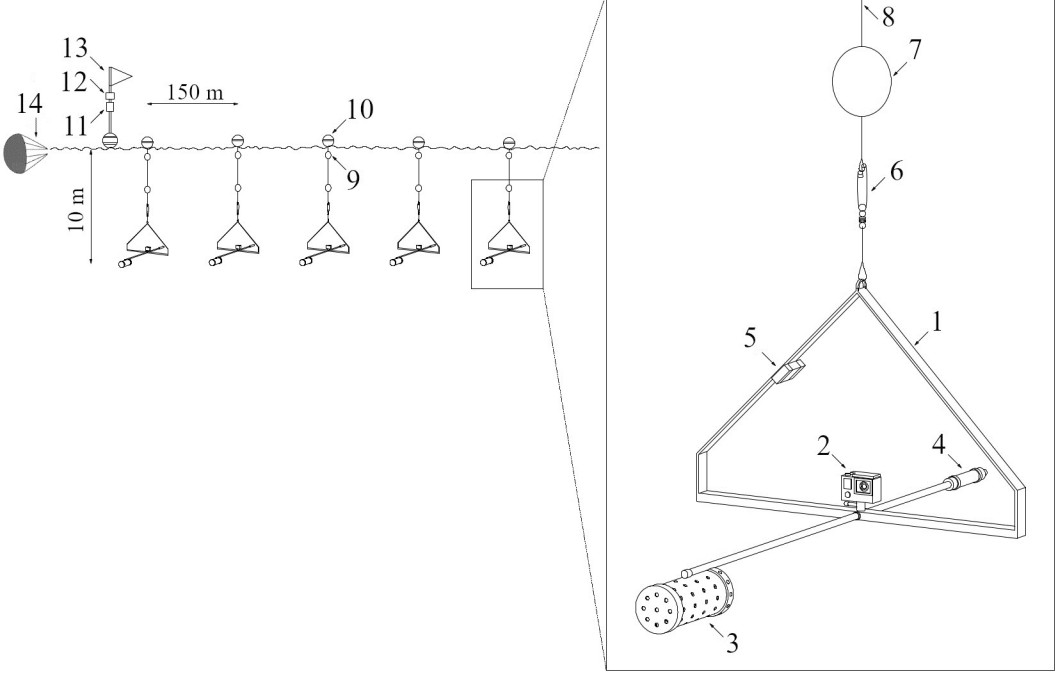

**Fig 2. Diagram of a shallow (10 m depth) drifting-pelagic Baited Remote Underwater Video Stations (BRUVS) deployment.** Numbers show closer details of a BRUVS unit: (1) Triangle-shaped stainless-steel frame; (2) GoPro camera; (3) Perforated PVC bait container; (4) Counterweight; (5) Temperature sensor; (6) Stainless steel tuna fishing swivel clip; (7) Small intermediate buoy; (8) 10 m or 25 m line; (9) Small surface buoy; (10) Big surface buoy; (11) Global Positioning System device (GPS); (12) Automatic Identification System (AIS) track device; (13) Flag; (14) Sea anchor. The configuration of the deep deployment is the same as the shallow one but BRUVS units are at 25m depth.

The drifting-pelagic BRUVS units were connected with a superficial tether line resembling a long-line system to maximize survey effort while minimizing the risk of loss and the amount of the tracking devices needed. Two long-lines with 5 drifting-pelagic BRUVS separated 150 meters each were simultaneously deployed several times at each seamount during the expedition (Fig 2), Hereon, we refer to each long-line with 5 connected BRUVS units as one deployment. To assess differences in species detectability between depth levels, simultaneous deployments were suspended at 10 m (shallow) and 25 m (deep) (Fig 2). To assess the effect of time of day on species richness and abundance, deployments were conducted during morning (6:00–10:00 am) and afternoon (1:00–3:00 pm) at each seamount. All deployments entered the water upstream of each seamount to ensure that the sound of the vessel engine was not present when the units drifted over the sampling sites. A sea anchor was used to stabilize deployments and avoid entanglements (Fig 2). The average distance left among simultaneous deployments was 1.6 ± 1.04 km (mean ± SD).

Differences in temperature between shallow and deep deployments per seamount were obtained attaching a temperature datalogger (ONSET Hobo Pendant® UA) to each BRUVS unit. The drifting distance of each deployment was measured fitting each deployment with a Global Positioning System device (GPS). An Automatic Identification System track device (AIS) was also attached to each deployment to capture its position while drifting with the current (Fig 2). Missing temperature and drifting distance values for some BRUVS (8% and 6.7% respectively) were obtained calculating the average of those values between the two nearest stations. Mean concentrations of chlorophyll-a around each BRUVS unit (spatial resolution of

4.64km) were gathered by the Moderate Resolution Imaging Spectroradiometer (MODIS) from NASA's Goddard Space Flight Center and the Ocean Biology Processing Group. The data was extracted using the ERDDAP open-source data server with the *rxtracto* function from the *rerdappXtracto* package (R software v 3.6.3). Additional information such as date, time, location and duration of each deployment was recorded on the field.

## Video and data analysis

The software EventMeasure (SeaGIS®) was used to analyze the video footage and calculate the relative abundance of each observed species. Relative abundance was defined as the maximum number of individuals from each species observed in a single frame of the video (MaxN). The MaxN is a conservative estimate of the total number of individuals present in the deployment area because it avoids repeat counts of animals reentering the field of view and because only a proportion of the species present in the deployment area will positively respond to the bait plume by entering the camera's field of view [63,64]. Species were identified to the lowest taxonomic level possible using local identification guides and expert knowledge if required. Species were classified in ecological groups according to taxonomy and reported common sizes [4] into elasmobranchs, large teleosts (species with common total length > 1m), small teleosts (species with common total length < 1m), dolphins and turtles. For some graphical representations, dolphins and sea turtles were pooled together as "Other LPS" since only one species from each group was observed. Small teleost richness and abundance data were described, but not included in the statistical comparisons since their higher abundance could mask the distribution and abundance patterns of LPS which are the group of interest of this study. For statistical purposes, LPS were pooled together.

Differences in sampling effort were standardized across deployments and seamounts by dividing MaxN of each species obtained per BRUVS unit by soak time (effective time of BRUVS recording), expressed as MaxN hr$^{-1}$. The accumulation rate of MaxN for all organisms and for LPS increased continuously with soaktime without reaching an inflection point (S1 Fig), allowing us to use MaxN hr$^{-1}$ without underestimating relative abundance. Considering that BRUVS from the same long-line were only separated 150 m from each other, we considered each deployment (one long-line with 5 connected BRUVS) as an independent replicate. Therefore, the MaxN hr$^{-1}$ considered for each species was the maximum value recorded among the five connected BRUVS of the same deployment. To compare the relative abundance across ecological groups and seamounts, the MaxN hr$^{-1}$ of each group per deployment was summed. Temperature, chlorophyll-a, latitude, longitude and soak time values were averaged among the five connected BRUVS of each deployment (S1 Data). Missing temperature and distance values for some BRUVS (8% and 6.7%, respectively) were obtained calculating the average of those values between the two nearest stations. Measures of dispersion of observed means were reported as mean ± standard deviation (SD).

Cumulative species richness curves were used to examine temporal accumulation of LPS species per group by deployment and soak time. The order in which species were analyzed was randomized 999 times and the cumulative number of new species per deployment was counted for each randomization. The function *specaccum* with "random" method from the vegan statistical package (R software v 3.6.3) was used to calculate the rate of species accumulation. The same function but with "*rarefaction*" method was used to calculate the rate of MaxN accumulation per soak time.

Poisson and Negative binomial generalized linear models (GLM) were used to examine relationships between a set of predictors and LPS richness and relative abundance (MaxN). A Negative binomial model was used as an alternative of a Poisson model when overdispersion

was detected (variance larger than the mean) [65]. Overdispersion was assessed using Pearson residuals, where a score of close to 1 is indicative of a lack of over-dispersion [65]. Since soak time differed per deployment, the log transformed values of soaktime (hours) was used as an offset for all models [65,66]. The categorical predictors selected include i) time of day (morning: 6:00–10:00 am; afternoon: 1:00–3:00 pm), since it might affect the vertical movement of pelagic species in offshore habitats [67–70]; ii) camera depth level (shallow: 10 m; deep: 25 m), since changes in water temperature and dissolved oxygen are often associated to shallow thermoclines (often at 25 m) in the ETP, and thus may be an important predictor of LPS distribution [54,71]; and iii) seamount depth based on summit depth level (shallow: < 400 m; deep: > 400 m), since marine predators in other regions have shown a higher tendency to be associated with features shallower than 400 m [42,57]. The continuous predictor variables included in the model were i) drifting distance of each deployment during the soaktime (0.8–8.9 km) as a proxy of current intensity, since this parameter could influence the abundance of hammerheads and other shark species [13,16,72]; ii) MPA distance defined as the minimum distance of each seamount to the boundaries of the nearest MPA (0–275.5 km), either Cocos or Galapagos Islands, both centers of marine diversity and abundance of LPS in the ETP [13,22,73,74]; iii) mean water temperature at the camera depth level (22.8–29.7 ˚C), an important determinant of LPS distribution [75,76]; and iv) mean chorophyll-a concentration (0.13–0.31 mg m$^{-3}$), an indicator of primary productivity and available trophic energy [32,75]. As visibility was always between 10 and 15 m in all samples, this parameter was not considered as an influencing factor when making comparisons between replicates. The Variance Inflation Factor (VIF) did not show signs of correlation among continuous predictor variables (VIF < 3) and therefore, all predictor variables were included in the full model for LPS richness and LPS relative abundance [77]. The distribution of the response and predictor variables was examined prior to perform models using diagnostic plots such as histograms, cleveland plots, pairplots and boxplots [65]. Model selection was based on the small sample-corrected Akaike's information criterion (AICc). This approach has been suggested as a useful option for small samples where the ratio of observations to model parameters is low (e.g $N/K < 40$) [78,79]. Models were ranked based on minimum AICc, detailing changes in AICc with respect to the top ranked model (ΔAICc) and model weights (wAICc) [80,81]. Models with values of ΔAICc ≤ 2 were presented, since values within this threshold can have similar explanatory power [81,82]. Model weights were computed as a measure of each model's strength of evidence where the smaller the wAICc, the lower probability the model is true [79]. The cumulative wAICc was used to identify a 95% confidence set of models and to measure the relative importance of each variable [65]. The larger the sum of the weight value ($\Sigma w_i$), the more important the variable is relative to the other variables [65,83]. Residual deviance and GLM diagnostic plots of standard residuals were used to evaluate the goodness of fit of the resulting models and to determine wheatear models assumptions were met [65]. The libraries *psych*, *car*, *pscl*, *MuMIn*, *MASS* and *lmtest* (R software v 3.6.3) were used to examine data and to perform model selection.

## Results

A total of 347.5 hours of video footage from 32 deployments (150 BRUVS units) were recorded in 9 seamounts along the Cocos Ridge (S1 Data). Although we aimed to conduct 4 deployments (20 BRUVS units) per seamount, due to logistic limitations, equipment losses or video failures, the total survey effort per seamount ranged from 11.7 to 55.1 hours (mean ± SD: 32.5 ± 13.3 hours) (Table 1).

**Table 1. Survey effort, location, depth and environmental data associated with seamounts surveyed along the Cocos Ridge.**

| Seamounts | D (N) | Hrs | Lat | Long | Dist (km) | Depth (m) | Temp (˚C) | Drift (km h⁻¹) | Chl-a (mg m³) |
|---|---|---|---|---|---|---|---|---|---|
| *Gemelas 2* | 2 (9) | 17.5 | 5.069 | -87.634 | 79 | -172 | 27.5 | 0.65 | 0.18 |
| *Gemelas 1* | 4 (20) | 39.9 | 4.985 | -87.401 | 70 | -198 | 28.5 | 0.86 | 0.21 |
| **East Cocos** | 4 (20) | 31.3 | 4.600 | -86.720 | 109 | -775 | 29.1 | 0.97 | 0.16 |
| *West Cocos* | 4 (20) | 55.1 | 5.467 | -88.533 | 162 | -283 | 29.2 | 0.47 | 0.13 |
| Medina 3 | 4 (18) | 26.5 | 3.334 | -88.268 | 276 | -445 | 27.9 | 2.19 | 0.21 |
| Medina 2 | 4 (15) | 38.4 | 2.990 | -88.050 | 324 | -688 | 27 | 3.08 | 0.28 |
| Medina 1 | 4 (19) | 29.8 | 3.235 | -88.935 | 327 | -818 | 27.4 | 1.91 | 0.31 |
| *Paramount* | 4 (19) | 43.1 | 3.349 | -90.781 | 227 | -188 | 27.9 | 1.11 | 0.14 |
| *NW Darwin* | 2 (10) | 11.7 | 1.881 | -92.134 | 27 | -1200 | 24.6 | 1.95 | 0.20 |

D—number of deployments; N—number of valid BRUVS units; Hrs—recorded hours with baited remote underwater video stations (BRUVS); Dist (km)—minimum distance to closest MPA (Cocos or Galapagos Islands); Depth (m)—depth of seamount summit; Temp—mean temperature (˚C); Drift—mean drifting speed (km h⁻¹) of pelagic-BRUVS deployments; Chl-a—mean concentration of chlorophyll-a (mg m⁻³). Seamounts are ordered with increasing distance from Cocos Island. Shallow seamounts (depth < 400m) are shown in italics.

## Richness and abundance

Our cameras detected LPS on 90.6% of all deployments (n = 32) or 48% of individual BRUVS (n = 150), with the number of species per deployment ranging from 1 to 6 (mean ± SD: 1.8 ± 1.3 species). Twenty-one species were identified from BRUVS footage, from which 13 species (62%) were LPS, including 6 teleosts (4 families), 3 sharks (3 families), 2 pelagic rays (2 families), 1 dolphin and 1 sea turtle (Table 2, Fig 3). Some species were only identified at the genus level (*Mobula* spp. and *Decaptreurs* spp.). Six species of small teleost from 4 families were also recorded (Table 2). Approximately, 56% of all small teleost sighted could not be identified due to their small size and/or low video resolution, hereafter referred to as "unidentified". Of all LPS identified to species level (n = 12), 33% were threatened species (n = 4), 58% were non threatened (n = 7) and only 9% (n = 1) were not evaluated based on current assessments from the International Union for the Conservation of Nature (IUCN) [84] (Table 2). All small teleosts recorded were classified as either Least Concern or Not Evaluated (Table 2).

Small teleosts were the most abundant (14.1 ± 59.7 MaxN hr⁻¹) and commonly sighted (94% of all deployments) group. Large teleosts and elasmobranchs occurred at 59% and 50% of all deployments, respectively, and showed similar relative abundances (large teleosts: 2.9 ± 4.4 MaxN hr⁻¹; elasmobranchs: 3.3 ± 6.1 MaxN hr⁻¹). The least sighted and abundant groups were dolphins, occurring at 28% of all deployments (1.8 ± 1.7 MaxN hr⁻¹), and sea turtles, occurring at 3.1% of all deployments with only 1 individual detected (0.4 MaxN hr⁻¹).

The scalloped hammerhead shark (*Sphyrna lewini*) dominated the elasmobranch group, representing 91.5% of the group's MaxN. Furthermore, this species showed the highest frequency of observation among all elasmobranchs (34.4% of all deployments) (Table 2). The common dolphinfish or mahi-mahi (*Coryphaena hippurus*) was the most abundant large-bodied teleost (91.8% of the group's MaxN) and the most frequently observed among all large-bodied teleosts (43.8% of all deployments) (Table 2). Among the small teleosts, the green jack (*Caranx caballus*) was the most abundant species (79.6% of the group's MaxN) and the pilot fish (*Naucrates ductor*) was the most frequently observed species (50% of all deployments) (Table 2). The bottlenose dolphin (*Tursiops truncatus*) and the black sea turtle (*Chelonia mydas)* were the only species of dolphins and sea turtles, respectively (Table 2). Unidentified species were not considered in the ranking since their associated values were summed across several unidentified species.

**Table 2. Summary of occurrence, relative abundance (MaxN and MaxN hr$^{-1}$) and conservation status of all species classified by groups recorded on baited remote underwater video stations (BRUVS) along Cocos Ridge seamounts.**

| EG/Family | Species | Common name | D | % D | MaxN | | MaxN hr$^{-1}$ (mean ± SD) | Rank | Status |
|---|---|---|---|---|---|---|---|---|---|
| | | | | | max | sum | | | |
| **Elasmobranchs** | | | | | | | | | |
| Sphyrnidae | *Sphyrna lewini* | Scalloped hammerhead | 11 | 34.4 | 60 | 162 | 6.3 ± 8 | H | CR |
| Carharhinidae | *Carcharhinus falciformis* | Silky shark | 5 | 15.6 | 2 | 7 | 0.6 ± 0.2 | M | VU |
| Alopiidae | *Alopias pelagicus* | Pelagic tresher shark | 3 | 9.4 | 1 | 3 | 0.4 | L | EN |
| Mobulidae | *Mobula spp.* | Mobula ray | 2 | 6.2 | 2 | 3 | 0.6 ± 0.3 | L | - |
| Dasyatidae | *Pteroplatytrygon violacea* | Pelagic stingray | 2 | 6.2 | 1 | 2 | 0.4 | L | LC |
| **Total** | | | **16** | **50** | - | **177** | **3.3 ± 6** | - | - |
| **Large teleosts** | | | | | | | | | |
| Coryphaenidae | *Coryphaena hippurus* | Common dolphinfish | 14 | 43.8 | 28 | 134 | 4.6 ± 5.1 | H | LC |
| Istiophoridae | *Kajikia audax* | Sriped marlin | 5 | 15.6 | 1 | 5 | 0.5 ± 0.2 | M | NT |
| | *Istiompax indica* | Black marlin | 1 | 3.1 | 1 | 1 | 0.4 | L | DD |
| | *Istiophorus platypterus* | Sailfish | 1 | 3.1 | 1 | 1 | 0.4 | L | LC |
| Scombridae | *Thunnus albacares* | Yellowfin tuna | 2 | 6.2 | 3 | 4 | 0.8 ± 0.5 | L | NT |
| Alepisauridae | *Alepisaurus ferox* | Long snouted lancetfish | 1 | 3.1 | 1 | 1 | 0.4 | L | LC |
| **Total** | | | **19** | **59** | - | **146** | **2.9 ± 4** | - | - |
| **Other large pelagic species** | | | | | | | | | |
| Delphinidae | *Tursiops truncatus* | Bottlenose dolphin | 9 | 28.1 | 10 | 33 | 1.8 ± 1.7 | M | LC |
| Cheloniidae | *Chelonia mydas* | Pacific black sea turtle | 1 | 3.1 | 1 | 1 | 0.4 | L | EN |
| **Total** | | | **9** | **28** | - | **34** | **1.7 ± 2** | | |
| **Small teleosts** | | | | | | | | | |
| Carangidae | *Naucrates ductor* | Pilot fish | 16 | 50 | 37 | 105 | 2.7 ± 3.4 | H | LC |
| | *Caranx caballus* | Green Jack | 4 | 12.5 | 500 | 553 | 55.4 ± 95.2 | M | LC |
| | *Seriola peruana* | Fortune jack | 3 | 9.4 | 12 | 19 | 2.6 ± 2.2 | M | LC |
| | *Seriola rivoliana* | Pacific amberjack | 1 | 3.1 | 6 | 6 | 2.5 | L | LC |
| | *Decapterus spp.* | Scads | 1 | 3.1 | 1 | 1 | 0.4 | L | NE |
| Monacanthidae | *Aluterus monoceros* | Unicorn filefish | 2 | 6.2 | 3 | 4 | 0.7 ± 0.5 | L | LC |
| Coryphaenidae | *Coryphaena equiselis* | Pompano dolphinfish | 1 | 3.1 | 4 | 4 | 1.8 | L | LC |
| Scombridae | *Sarda orientalis* | Striped bonito | 1 | 3.1 | 3 | 3 | 1.2 | L | LC |
| Unidentified | Unidentified | Unidentified | 31 | 96.9 | 1000 | 1301 | 17.9 ± 74.9 | H | - |
| **Total** | | | **30** | **94** | - | **1996** | **14.1 ± 59** | | |

Species are organized by families and ecological groups (EG). D: Number of BRUVS deployments with a species. Relative abundance is expressed as (i) MaxN: Maximum number of individuals of a species per deployment (one deployment with 5 connected BRUVS is considered an independent sample) and (ii) MaxN hr$^{-1}$: MaxN divided by soak time (hours). Species are ranked based on the frequency of occurrence: H—high (>30% of deployments); M—medium (10—30% of deployments); and L—low (<10% of deployments). Conservation status of each species (DD—data deficient, LC—least concern, NT—near threatened, VU—vulnerable, EN—endangered, and CR—critically endangered) is presented based on current IUCN Red List Assessments [84].

The slope of the species accumulation curve indicated that the number of species for all groups increased gradually with number of BRUVS deployments and soak time, with some differences found between groups (S2 Fig). Although still increasing, the species accumulation curve for small teleosts nearly reached an asymptote after 180 min of soak time, whereas elasmobranchs and other marine megafauna species showed signs of stabilization after 90 min of soak time. The curve for large teleosts continued to increase at a maximum soak time of 140 min. Species richness accumulation curves increased at a faster rate for small teleosts and elasmobranchs (sharp increase during the first 20 min), followed by large teleosts and ultimately by other LPS (dolphins and 1 sea turtle) at a much slower rate (S2 Fig).

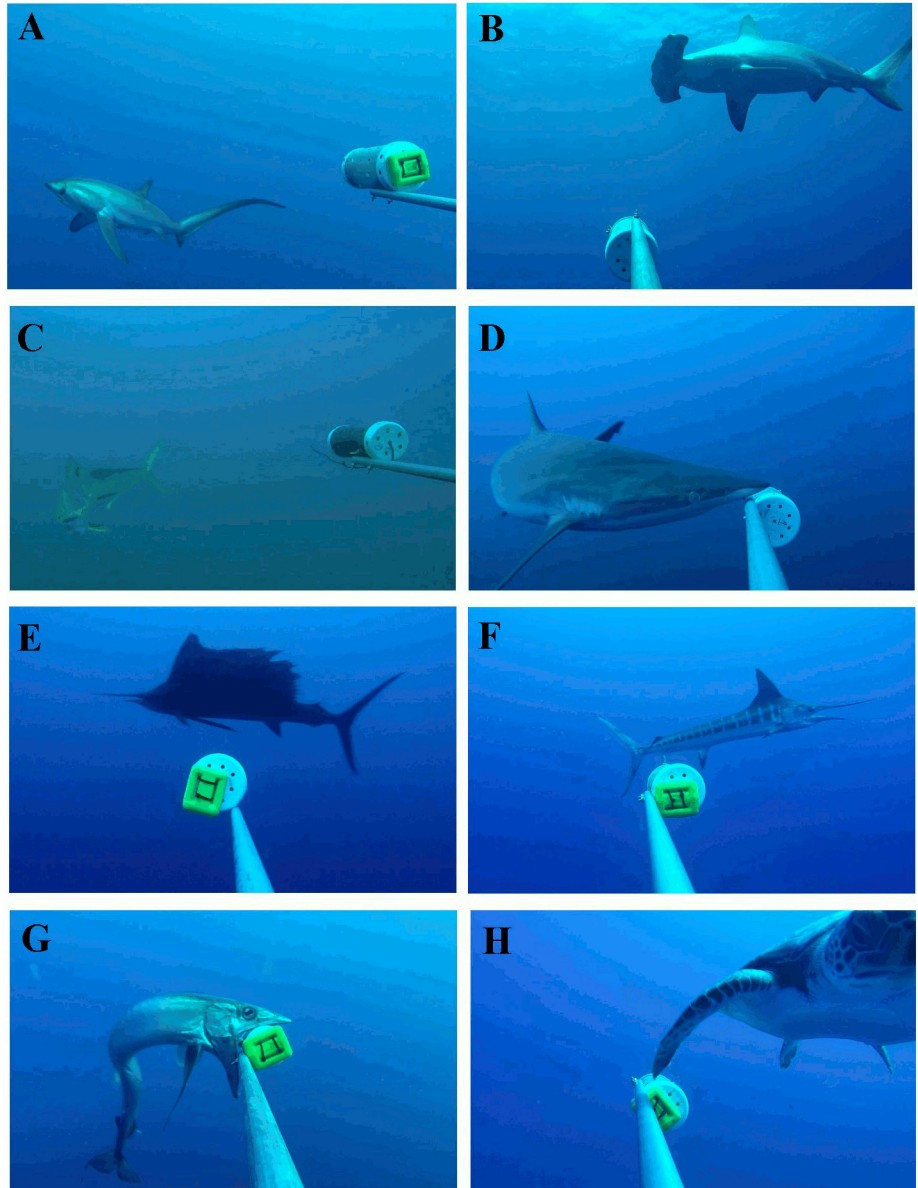

**Fig 3. Images of large pelagic species detected by baited remote underwater video stations (BRUVS).** A) Thresher shark *(Alopias pelagicus)*. B) Hammerhead shark *(Sphyrna lewini)*. C) Mahi-mahi *(Coryphaena hippurus)*. D) Silky shark *(Carcharhinus falciformis)*. E) Sail fish *(Istiophorus platypterus)*. F) Stripped marlin *(Kajikia audax)*. G) Lancet fish *(Alepisaurus ferox)*. D) Pacific black sea turtle *(Chelonia mydas)*.

## Spatial distribution across seamounts

The highest richness and relative abundance of LPS were found at West Cocos (3.8 ± 1.5 species; 12.8 ± 6.2 MaxN hr$^{-1}$) and Paramount (2.5 ± 1 species; 9.1 ± 9.9 MaxN hr$^{-1}$) seamounts, whereas the lowest values were found at Medina 3 (0.5 ± 0.6 species; 0.2 ± 0.3 MaxN hr$^{-1}$) and NW Darwin (0.5 ± 0.7 species; 0.2 ± 0.4 MaxN hr$^{-1}$) seamounts (Fig 4; S1 Table). The number of LPS ranged from 1 species at NW Darwin to 8 species at West Cocos (Fig 4A; S1 Table). Elasmobranchs were detected at all seamounts except for Medina 1 and NW Darwin, whereas large teleosts were not detected at Paramount and Las Gemelas 2. Other LPS such as dolphins

**A.**

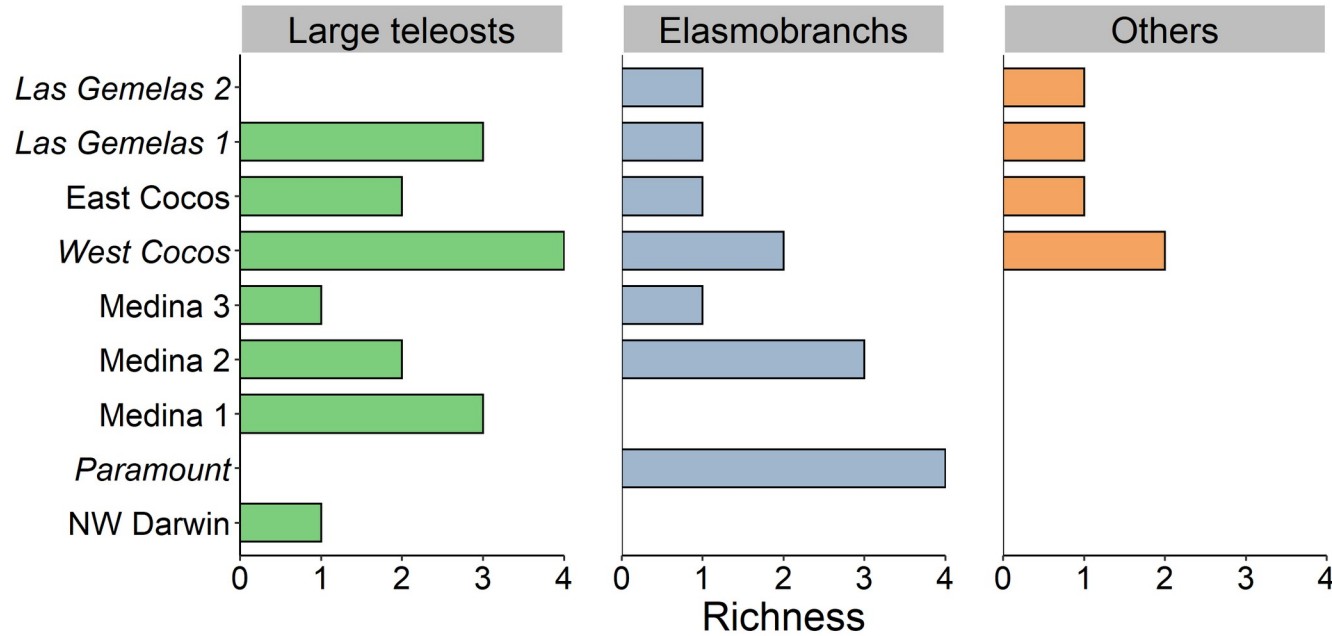

**B.**

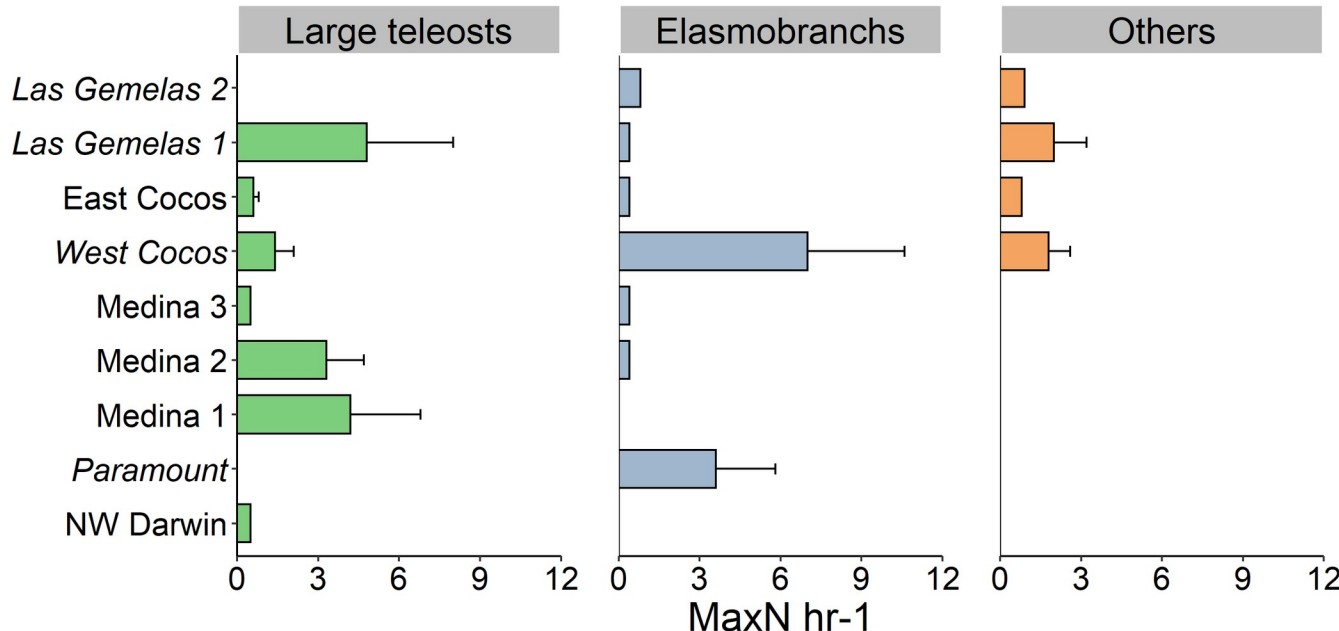

**Fig 4.** Comparison of (A) species richness and (B) relative abundance (MaxN hr$^{-1}$) among ecological groups. Seamounts are ordered from bottom to top according to the distance from Galapagos Islands (bottom) to Cocos Island (top). Error bars represent the 95% confidence intervals. Shallow seamounts (<400 m) are shown in italics.

and sea turtles were only reported at seamounts close to Cocos Island including Las Gemelas seamounts, West Cocos and East Cocos (Fig 4A). The highest elasmobranch richness was reported in Paramount, whereas the highest richness of large teleosts were reported at West Cocos (Fig 4A).

Elasmobranchs were more abundant at West Cocos and Paramount seamounts (Fig 4B), where large schools of *S. lewini* of up to 40 and 60 individuals were observed in a single deployment, respectively (S1 Table). Few individuals of *S. lewini* were also detected in Medina 3 and Gemelas 2 (S1 Table). Large teleosts were more abundant at Las Gemelas 1 and Medina 1 (Fig 4B), where our BRUVS reported schools of 27 and 28 individuals of *C. hippurus* in a single deployment, respectively (S1 Table). However, the highest abundance of *T. albacares* and billfishes (*Istiompax indica*, *Istiophorus platypterus* and *Kajikia audax*) was reported at West Cocos (S1 Table). Dolphins and sea turtles were more abundant at West Cocos and Las Gemelas 1 (Fig 4B), where our BRUVS recorded groups of up to 6 and 10 *T. truncatus*, respectively (S1 Table). The only individual of *C. mydas* was reported at West Cocos (S1 Table).

## Drivers of large pelagic species

Based on the GLMs examined, LPS richness and relative abundance (MaxN) followed a Poisson and a Negative Binomial distribution, respectively. The large number of models in the 95% confidence set of models ($\Sigma w_i$ = 0.95) for both LPS richness (66 models; S2 Table) and LPS relative abundance (27 models; S3 Table) indicated a considerable model uncertainty regarding the identity of the best approximating model. The low wAICc of the top ranked models ($\Delta$AIC $\leq$ 2) for both response variables is another indicative of model uncertainty (Table 3). The Poison GLM with the highest wAICc included seamount depth and minimum distance to MPA as predictor variables of LPS richness (Table 3). However, seamount depth was the only variable with a significant effect over LPS richness (Table 3), and the variable with the highest relative importance ($\Sigma w_i$ = 0.89) from all variables examined (S2 Table). Although the minimum distance to a MPA was not a significant variable for LPS richness (Table 3), it showed a moderately higher relative importance ($\Sigma w_i$ = 0.52) compared to the other variables ($\Sigma w_i$ < 0.3). Richness of LPS was significantly higher at shallow seamounts relative to deeper ones (Fig 5A) and increased with increasing distance to nearest MPA in both shallow and deep seamounts (Fig 5B). Diagnostic plots of standard residuals from the first top ranked model indicated that model fit for LPS richness was appropriate with respect to heteroscedasticity and normality of residuals (S3 Fig).

**Table 3. Comparison of optimal generalised linear models (GLM), using a Poisson and a Negative Binomial error distribution, of large pelagic species (LPS) richness and relative abundance (MaxN), respectively.**

| Richness LPS—Poisson GLM | RD | df | LL | AICc | ΔAICc | wAICc |
|---|---|---|---|---|---|---|
| Richness ~ **seamount depth** + MPA distance, offset = log (soaktime) | 15.8 | 3 | -37.35 | 81.56 | 0.00 | 0.13 |
| Richness ~ **seamount depth**, offset = log (soak time) | 18 | 2 | -38.97 | 82.35 | 0.79 | 0.09 |
| Richness ~ **seamount depth** + MPA distance + time, offset = log (soak time) | 14.9 | 4 | -36.88 | 83.25 | 1.69 | 0.06 |
| Richness ~ **seamount depth** + MPA distance + camera depth, offset = log (soak time) | 15.1 | 4 | -36.99 | 83.47 | 1.91 | 0.05 |
| **MaxN LPS—Negative Binomial GLM** | **RD** | **df** | **LL** | **AICc** | **ΔAICc** | **wAICc** |
| MaxN ~ **seamount depth** + MPA distance, offset = log (soak time) | 29.1 | 4 | -82.08 | 173.64 | 0.00 | 0.24 |
| MaxN ~ **seamount depth** + MPA distance + time, offset = log (soak time) | 28.8 | 5 | -81.41 | 175.13 | 1.49 | 0.11 |
| MaxN ~ **seamount depth** + MPA distance + chla, offset = log (soak time) | 29.1 | 5 | -81.65 | 175.61 | 1.98 | 0.09 |

Models presented are those with lowest values of the Akaike Information Criterion corrected for small sample sizes (AICc). Models are ranked by increasing AICc value. Predictors used at each model were: 1) time—time of deployment (morning 6—10 am/afternoon 1—3 pm); 2) camera depth—camera depth level (shallow 10m/deep 25m); 3) seamount depth—summit depth level (shallow < 400m/deep > 400m); 4) drift—drifting distance of each deployment during the effective recording time (soak time); 5) MPA distance—minimum distance of each seamount to the boundaries of Cocos or Galápagos Islands; 6) temp—mean temperature at the camera depth level and 7) chla—mean chorophyll-a concentration. Log of soak time (hours of effective recording) was used as an offset in the models. Residual deviance (RD), maximum Log Likelihood (LL), degrees of freedom (df), difference of AICc of a given model to the model with best fit (ΔAICc) and relative model probability expressed as AICc weight (wAICc) are shown for each model. Significant predictors are highlighted in bold ($\alpha$ = 0.05). Only models with $\Delta_i$ values $\leq$ 2 are presented.

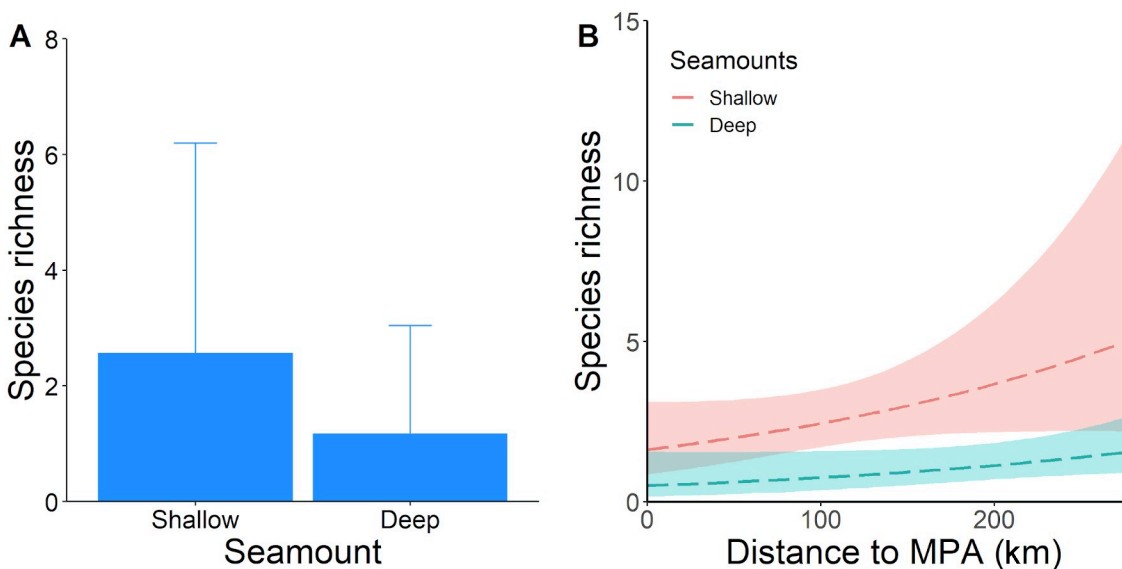

**Fig 5. Graphical results from the two top-ranked AICc generalized linear models for large pelagic species (LPS) richness.** A) Estimates of LPS richness (mean ± SD) at shallow (<400 m) and deep (>400 m) seamounts; B) Relationship between richness of LPS and distance to the nearest Marine Protected Area (Cocos or Galapagos Islands) at shallow (<400 m) and deep (>400 m) seamounts.

The Negative binomial GLM with the highest wAICc included seamount depth and minimum distance to a MPA as predictor variables of LPS relative abundance (Table 3). These were the only variables with a significant influence over LPS relative abundance (Table 3) and with the highest relative importance ($\Sigma w_i = 0.95$) compared to the other predictors (S3 Table). Relative abundance of LPS was higher at shallow than deep seamounts and increased with increasing distance from the nearest MPA (either Cocos or Galapagos Islands) (Fig 6). Diagnostic plots of standard residuals from the first top ranked model indicated that model fit for LPS relative abundance was satisfactory with respect to heteroscedasticity and normality of residuals (S4 Fig).

The models with the lowest residual deviance and within a $\Delta$AIC $\leq$ 2 from the top-ranked model for both LPS richness and LPS relative abundance included time of deployment in addition to seamount depth and minimum distance to MPA (Table 3). Afternoon deployments had higher species richness (S5 Fig) and relative abundance (S6 Fig) compared to morning deployments although the differences were not statistically significant. Time of deployment also presented the highest relative importance value among the non-significant predictor variables for both LPS richness and relative abundance. However, the differences among them were small (S3 Table). Other models within a $\Delta$AIC $\leq$ 2 from the top-ranked model also included camera depth level and chlorophyll-a as predictor variables for LPS richness and relative abundance, respectively (Table 3).

## Discussion

Our results suggest that Cocos Ridge seamounts connecting Cocos Island (Costa Rica) and the Galapagos Islands (Ecuador) are important aggregation sites for wide ranging marine species such as elasmobranchs, large teleosts, dolphins and sea turtles. Since these species are suffering severe population declines across the ETP due to fishing pressure [21,30,85–87], it is critical to identify and protect key habitats in open-oceans [2,3,8,42,88]. Our study showed that drifting pelagic BRUVS are an effective, accessible and non-extractive technique to monitor LPS in

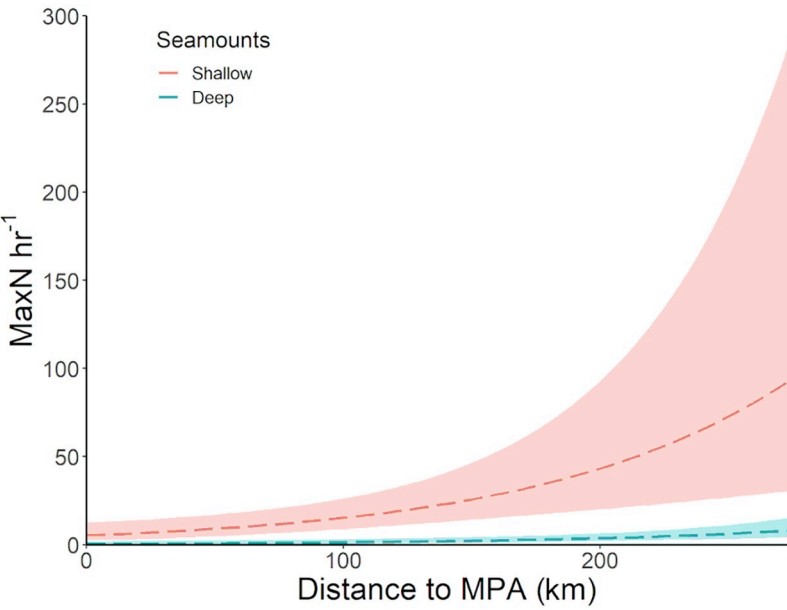

**Fig 6. Graphical results from the top-ranked AICc generalized linear model for large pelagic species (LPS) relative abundance (MaxN hr⁻¹).** Representation of the relationship between MaxN hr⁻¹ of LPS and distance to nearest MPA (Cocos or Galapagos Islands) at shallow (<400 m) and deep (>400 m) seamounts.

oceanic environments, highlighting its potential to guide management and conservation actions in the ETP.

## Large pelagic species at seamounts between Cocos and Galapagos Islands

The evidence of the seamount effect on aggregating pelagic fauna has been demonstrated in a numerous studies and for a wide variety of species including zooplankton [89,90], teleosts [57,91–94], sharks [42,95,96], dolphins [97] and sea birds [98]. Although we need a higher spatial and temporal replication to elucidate if LPS along Cocos Ridge are positively associated to seamounts presence, our BRUVS recorded LPS (swimming individually or in large schools) in the vicinity of all surveyed seamounts in an open ocean environment where species richness is naturally low [99]. These results favor the hypothesis that seamounts might serve as important stepping stones where LPS aggregate when moving between oceanic islands of the ETP [20,100]. However, it is important to recognize that our data was taken by the end of La Niña phenomenon, when the thermocline is usually shallower and chlorophyll-a levels are usually higher than normal [53], potentially increasing species detectability. Moreover, future research should consider deploying drifting-pelagic BRUVS at different distances from seamounts and across different seasons in order to elucidate the aggregation effect of these topographic features on LPS [57].

Our models identified the depth of seamount summit as the most significant driver for LPS richness and abundance, which were significantly higher at shallow seamounts (< 400 m) compared to deeper ones (> 400m). The identification of seamount depth as an important driver for LPS is consistent with previous studies, where seamounts shallower than 400 m and 500 m at the Azores Islands [8] and the Indo-Pacific region [42], respectively, showed a significant aggregation effect on predator species. Higher fishing catches have been also reported to occur at seamounts shallower than 400 m compared to deeper ones [90]. The most common

explanation given to this pattern is that shallower abrupt topographies accumulate larger zooplankton biomass over the summit (known as the Taylor column generation effect), providing sufficient food to maintain higher trophic levels and commercial fisheries [90,101]. There is also the possibility that our BRUVS located at a shallow depth from the surface failed at capturing pelagic life at deeper levels. Future studies deploying drifting-pelagic BRUVS at different depth levels (> 25 m) are necessary to evaluate how pelagic assemblages vary along depth gradients away from the surface [41,102,103].

Distance to nearest MPA was also a significant predictor for LPS abundance, which increased at increasing distances from the nearest MPA (Cocos or Galapagos Islands). Although this variable was not a significant predictor for LPS richness, it was present in the first ranked model with the lowest AICc and the highest wAIC. Additionally, this variable presented a high relative importance suggesting some degree of influence over LPS richness. Current literature suggests that MPAs have a beneficial spill-over effect over adjacent areas [104,105], and therefore, higher LPS richness and abundance are expected to occur at seamounts closer to Cocos and Galapagos Islands. Since all captured LPS move beyond the protection limits of Cocos and Galapagos Islands, these species are probably not receiving all the benefits from the MPAs, and therefore, their abundance might not be increasing in adjacent areas [106,107]. Our results may indicate that remote, shallow seamounts in the ETP, in the absence of other nearby topographic structures, represent important aggregation sites of LPS in the open ocean, whereas seamounts closer to MPAs might be sporadically used by LPS that are aggregating inside MPAs where they find more resources and better protection [13,25]. Another possible explanation is that greater fishing pressure surrounding the boundaries of MPAs in the ETP is negatively affecting LPS populations at closer seamounts [108,109]. However, it is important to recognize that our survey effort at Las Gemelas 1, Las Gemelas 2, and NW Darwin (closest seamounts to Cocos and Galapagos Islands respectively) was lower (2 deployments) than the rest of seamounts (4 deployments). Therefore, a higher survey effort would allow a better understanding of the effect that MPA proximity has on LPS abundance across seamounts.

Our results suggest that Paramount and West Cocos seamounts may play important roles as aggregation sites for LPS along the Cocos Ridge. Based on the results from our models, it is likely that the higher abundance and richness of LPS at these seamounts was partially explained by a combination of a shallower seamount summit with a high degree of isolation compared to the other seamounts that only met one of the two conditions (Table 1). Large aggregations of *S. lewini* were only found at Paramount and West Cocos and therefore we recommend a special emphasis on both seamounts for future studies and conservation planning in the region.

As a general rule of thumb, models with $\Delta_i$ values less than 2 are considered to be essentially as good as the best model [81,82]. Under this consideration, some of the non-significant predictor variables should be further examined as potential drivers of LPS richness and abundance in future studies. For example, time of deployment was included in the third and second ranked candidate models with $\Delta_i$ values $\leq 2$ for LPS richness and abundance, respectively. Our data showed a higher richness and abundance of LPS in afternoon deployments, although differences with morning deployments were not significant. Since the bottom trapping mechanism of migrating zooplankton accumulation over the seamounts occurs early in the morning [101] we would have expected to find a higher LPS richness and abundance in the morning deployments associated with a feeding behavior of LPS close to the surface [110]. However, most of our cameras were deployed few hours after dusk (S1 Data), when the trapped zooplankton might have already been consumed or descended to deeper levels [101]. Afternoon deployments instead, remained in the water until dawn when migrating zooplankton begin

the ascent towards the surface increasing the probability of detection of larger predators [103]. Since marine predators might use species-specific daily routines depending on prey distribution and environmental changes in order to optimize fitness [110–113], a statistical analysis at a group or species level could better explain the observed patterns.

Other non-significant predictor variables included in models with $\Delta_i$ values $\leq$ 2 where camera depth level in the fourth ranked candidate model for LPS richness and chlorophyll-a in the third ranked candidate model for LPS abundance. These parameters are known to influence abundance and distribution patterns of marine megafauna in pelagic environments [75,103,114,115]. However, in this study both variables showed low values of relative importance and the models were they were included had the lowest wAICc among the best candidate models. Future studies are needed to better understand the effect of these variables on LPS of the ETP. Dissolved oxygen [54], ocean currents [64,90], prey availability [57], Earth magnetic field [116,117], and fishing effort [42] may also pose a significant influence on the use of seamounts by LPS, and therefore, should be considered as potential drivers in future studies.

## Drifting-pelagic BRUVS to survey pelagic ecosystems in the ETP

Studies where drifting-pelagic BRUVS have been used in fully pelagic ecosystems are scarce [41,42]. Despite the short-term nature of our study (150 BRUVS or 32 deployments in 9 days of survey), our drifting-pelagic BRUVS showed a high frequency of LPS detection (90.6% of all deployments or 48% of all BRUVS) compared to similar studies in open-water ecosystems of Australia (n = 51 BRUVS) [41] and the Indo-Pacific region (n = 1041 BRUVS) [42], where LPS detection was 27% and 34% respectively. These results show a promising potential of the drifting-pelagic BRUVS to survey LPS in pelagic environments of the ETP.

Scientific efforts in the region have already demonstrated that sharks and other LPS move between Cocos and Galapagos Islands [16–19] suggesting the existence of a marine corridor among both MPAs [20]. However, this is the first fishery-independent study providing valuable insights on the diversity and community structure of LPS outside the protection limits of both MPAs. Fishery-dependent data in the region has provided information on distribution, diversity, effect of environmental drivers and size-structure of many of LPS recorded by our BRUVS [30–32,87]. However, sampling effort and location from these studies depend on fishing activities, and therefore, they do not provide specific information from sites of special biological interest such as Cocos Ridge seamounts. Instead, drifting-pelagic BRUVS provide a cost-effective and non-destructive alternative to monitor marine communities *in situ*, and therefore, have the potential to generate novel data at specific sites of interest along unexplored open-water ecosystems. For example, our BRUVS captured the first visual evidence of the schooling behavior of *S. lewini* at open water ecosystems of the ETP, making this critically endangered species even more vulnerable to pelagic-longline and purse-seine fisheries [30,118,119].

Although baited camera surveys have been positively validated against extractive methods such as trawling and longlines [43,120], the use of BRUVS also have several limitations [40,121]. For example, bait plume dispersion is complex and dynamic which makes surveyed area unknown [64]. Also, bait responses behaviors may lead to sampling biases towards larger mobile species, yet in this scenario, they were the target group of our study. Since our BRUVS were deployed in offshore clear water (> 15 meters), visibility did not compromise our data. Although our cameras recorded under a wide field of view (220 degrees), there is still a portion of the surrounding area that was not sampled, and therefore counts of relative abundance could be underestimated. This problem can be solved by using 360˚ cameras that allow a full field of view around each BRUVS, but it will also increase surveying costs, analysis time and

limit comparisons with previous studies. Despite the above mentioned limitations, drifting-pelagic BRUVS generate relevant ecological data of apex predator guilds that are typically cryptic, increasingly exposed to anthropogenic mortality and of high conservation and commercial value without posing a threat to targeted species [40].

Given the novelty of this study, our results can be used as a first approach to guide future studies using drifting-pelagic BRUVS in the ETP. For example, the species accumulation curves showed that at least 90 min were needed to record a representative sample of apex predators such as elasmobranchs and cetaceans, whereas soak times up to 180 minutes were necessary to increase species detection of teleosts (S2 Fig). These results coincide with those reported by [121] where the slope of the curve for pelagic species in Western Australia was reduced after 90 minutes of soak time with a trend still increasing at a reduced rate at 180 minutes. In consideration of the uncertainty associated with bait plume dispersal and sampling area during BRUVS studies [40], we recommend considering each deployment (with at least three BRUVS units each at a minimum distance of 200 m) as an independent replicate. To reduce potential biases associated with pseudo-replication we recommend leaving at minimum of 1 km as a conservative distance between independent drifting-pelagic BRUVS deployments.

## Conclusion

Our study represents the first attempt at characterizing the spatial distribution and relative abundance of LPS near seamounts along the Cocos Ridge, providing a first insight on how pelagic communities are structured outside the protection limits of two of the most important oceanic MPAs of the ETP. Our results show that shallow seamounts (<400 m) located at greater distances from MPAs may represent ecologically important refuges and foraging sites for LPS in the ETP, particularly along the Cocos Ridge. Future research is necessary to assess the potential of seamounts as high-priority conservation areas to prevent threatened species from further declines. Our results show a promising potential of the drifting-pelagic BRUVS to survey LPS in pelagic ecosystems of the ETP and elsewhere. This study might serve as an important reference for future studies in the region using this technique.

## Supporting information

**S1 Fig. Relative abundance (MaxN) accumulation curve per soak time of A) all organisms and B) only large pelagic species (LPS).** Soak time is defined as the effective recording time of Baited Remote Underwater Video Stations (BRUVS). Each deployment with 5 connected BRUVS is treated as an independent sample.
(TIF)

**S2 Fig. Species accumulation curves per ecological group.** (A) Species richness of each ecological group over cumulative soak time in minutes. Soak time is defined as the effective recording time of Baited Remote Underwater Video Stations (BRUVS). (B) Species richness of each ecological group over the number of BRUVS deployments. Each deployment with five connected BRUVS is treated as an independent sample. Shade colors represent 95% confidence intervals.
(TIF)

**S3 Fig. Diagnostic plots of standard residuals from the first top ranked Poisson GLM for large pelagic species richness.** The model is presented in Table 3.
(TIF)

**S4 Fig. Diagnostic plots of standard residuals from the first top ranked Negative Binomial GLM for large pelagic species relative abundance.** The model is presented in Table 3.
(TIF)

**S5 Fig. Graphical results from the third ranked Poisson generalized linear model for large pelagic species (LPS) richness.** Representation of the relationship between richness of LPS and distance to nearest MPA (Cocos or Galapagos Islands) at shallow (<400 m) and deep (>400 m) seamounts during morning and afternoon deployments.
(TIF)

**S6 Fig. Graphical results from the second ranked Negative Binomial generalized linear model for large pelagic species (LPS) relative abundance.** Representation of the relationship between relative abundance of LPS and distance to nearest MPA (Cocos or Galapagos Islands) at shallow (<400 m) and deep (>400 m) seamounts during morning and afternoon deployments.
(TIF)

**S1 Table. Relative abundance (MaxN hr$^{-1}$) and richness of large pelagic species (LPS) by seamount.** Relative abundance is expressed as MaxN hr$^{-1}$ (maximum number of individuals of a species recorded on a single deployment standardized by soak time). MaxN hr$^{-1}$ Species are abreviated as: AF) *Alepisaurus ferox*, AP) *Alopias pelagicus*, CF) *Carcharhinus falciformis*, CM) *Chelonia mydas*, CH) *Coryphaena hippurus*, II) *Istiompax indica*, IP) *Istiophorus platypterus*, KA) *Kajikia audax*, M) *Mobula* spp., PV) *Pteroplatytrygon violacea*, SL) *Sphyrna lewini*, TA) *Thunnus albacares* and TT) *Trusiops truncatus*. The MaxN hr$^{-1}$ at a group level for the small teleost species (ST) is also presented. The sum and the mean of LPS MaxN hr$^{-1}$ and species richness are also presented per seamount. Seamounts are ordered according to the sum of MaxN hr$^{-1}$. Shallow seamounts (<400 m) are shown in italics.
(DOCX)

**S2 Table. Model rankings of Poisson GLM for large pelagic species richness in the 95% confidence set of models and the relative importance of each predictor variable.** Models are ranked based on minimum Akaike Information Criterion corrected for small sample sizes (AICc). Predictors used at each model were: 1) time—time of deployment (morning 6—10 am/afternoon 1—3 pm); 2) camera depth—camera depth level (shallow 10m/deep 25m; 3) seamount depth—summit depth level (shallow < 400m/deep > 400m); 4) drift—drifting distance of each deployment during the effective recording time (soak time); 5) MPA distance—minimum distance of each seamount to the boundaries of Cocos or Galápagos Islands; 6) temp—mean temperature at the camera depth level and 7) chla—mean chorophyll-a concentration. Log of soak time (hours of effective recording) was used as an offset in the models. Degrees of freedom (df), maximum Log Likelihood (LL), degrees of freedom (df), difference of AICc of a given model to the model with best fit (ΔAICc) and relative model probability expressed as AICc weight (wAICc) are shown for each model. The sum of the wAICc was used to calculate the relative importance of each variable in the 95% confidence set of models following [79,122]. The function dredge from the MuMin Package in R version 3.6.3 was used to obtain this table.
(XLSX)

**S3 Table. Model rankings of Negative Binomial GLM for large pelagic species relative abundance in the 95% confidence set of models and the relative importance of each predictor variable.** Models are ranked based on minimum Akaike Information Criterion corrected for small sample sizes (AICc). Predictors used at each model were: 1) time—time of

deployment (morning 6—10 am/afternoon 1—3 pm); 2) camera depth—camera depth level (shallow 10m/deep 25m; 3) seamount depth—summit depth level (shallow < 400m/ deep > 400m); 4) drift—drifting distance of each deployment during the effective recording time (soak time); 5) MPA distance—minimum distance of each seamount to the boundaries of Cocos or Galápagos Islands; 6) temp—mean temperature at the camera depth level and 7) chla —mean chorophyll-a concentration. Log of soak time (hours of effective recording) was used as an offset in the models. Degrees of freedom (df), maximum Log Likelihood (LL), degrees of freedom (df), difference of AICc of a given model to the model with best fit (ΔAICc) and relative model probability expressed as AICc weight (wAICc) are shown for each model. The sum of the wAICc was used to calculate the relative importance of each variable in the 95% confidence set of models following [79,122]. The function dredge from the MuMin Package in R version 3.6.3 was used to obtain this table.
(XLSX)

**S1 Data. Raw data of environmental and biological information associated to each deployment.**
(CSV)

## Acknowledgments

This study would not have been possible without the support and collaboration of the Área de Conservación Marina Cocos, the Galapagos Science Center and Dirección del Parque Nacional Galapagos. We would like to thank Isaac Chaves, Tatiana Araya and Jorge Valerio who supported us with the video analysis. We also would like to thank all the crew from the Plan B vessel, the research boat used during the expedition, for their constant support during the field work. We are also grateful with Ignasi Agustí Fuster for his collaboration on the Baited Remote Underwater Video Station diagram presented on this paper.

## Author Contributions

**Conceptualization:** Marta Cambra, Frida Lara-Lizardi, César Peñaherrera-Palma, Alex Hearn, James T. Ketchum, Patricia Zarate, Carlos Chacón, Jenifer Suárez-Moncada, Esteban Herrera, Mario Espinoza.

**Data curation:** Marta Cambra, Frida Lara-Lizardi, Patricia Zarate, Carlos Chacón, Jenifer Suárez-Moncada, Esteban Herrera, Mario Espinoza.

**Formal analysis:** Marta Cambra, Frida Lara-Lizardi, Mario Espinoza.

**Funding acquisition:** Carlos Chacón.

**Investigation:** Marta Cambra, Frida Lara-Lizardi, César Peñaherrera-Palma, Alex Hearn, Patricia Zarate, Carlos Chacón, Jenifer Suárez-Moncada, Esteban Herrera, Mario Espinoza.

**Methodology:** Marta Cambra, Frida Lara-Lizardi, César Peñaherrera-Palma, Alex Hearn, James T. Ketchum, Mario Espinoza.

**Project administration:** Alex Hearn, Carlos Chacón.

**Supervision:** César Peñaherrera-Palma, Alex Hearn, Carlos Chacón, Mario Espinoza.

**Visualization:** Marta Cambra.

**Writing – original draft:** Marta Cambra, Mario Espinoza.

**Writing – review & editing:** Marta Cambra, Frida Lara-Lizardi, César Peñaherrera-Palma, Alex Hearn, James T. Ketchum, Mario Espinoza.

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
