## [Decision Letter · Decision Letter 0]

21 Jan 2021

PONE-D-20-38469

A first assessment of the distribution and abundance of large pelagic species at Cocos Ridge seamounts (Eastern Tropical Pacific) using drifting pelagic baited remote cameras

PLOS ONE

Dear Mrs Cambra,

Thank you for submitting your manuscript “A first assessment of the distribution and abundance of large pelagic species at Cocos Ridge seamounts (Eastern Tropical Pacific) using drifting pelagic baited remote cameras” to PLOS ONE. I have now received feedback from two experts in the field. As you can see below, they both found the study of interest, but also highlighted some major issues (especially reviewer 1) that prevent publication of this work in its current form. They provided constructive comments that will help you to improve your paper.

As a result, I would consider a resubmission thoroughly addressing the comments made by these reviews.

Please note that Reviewer 1 made some comments directly on the PDF.

We look forward to receiving your revised manuscript.

Kind regards,

Johann Mourier, Ph.D.

Academic Editor

PLOS ONE

Additional Editor Comments:

Dear Mrs Cambra,

Thank you for submitting your manuscript “A first assessment of the distribution and abundance of large pelagic species at Cocos Ridge seamounts (Eastern Tropical Pacific) using drifting pelagic baited remote cameras” to PLOS ONE. I have now received feedback from two experts in the field. As you can see below, they both found the study of interest, but also highlighted some major issues (especially reviewer 1) that prevent publication of this work in its current form. They provided constructive comments that will help you to improve your paper.

As a result, I would consider a resubmission thoroughly addressing the comments made by these reviews.

With kind regards,

Johann Mourier

Journal Requirements:

4. We note that Figure 1 in your submission contain map images which may be copyrighted. All PLOS content is published under the Creative Commons Attribution License (CC BY 4.0), which means that the manuscript, images, and Supporting Information files will be freely available online, and any third party is permitted to access, download, copy, distribute, and use these materials in any way, even commercially, with proper attribution. For these reasons, we cannot publish previously copyrighted maps or satellite images created using proprietary data, such as Google software (Google Maps, Street View, and Earth). For more information, see our copyright guidelines: http://journals.plos.org/plosone/s/licenses-and-copyright.

4.1.    You may seek permission from the original copyright holder of Figure 1 to publish the content specifically under the CC BY 4.0 license. 

4.2.    If you are unable to obtain permission from the original copyright holder to publish these figures under the CC BY 4.0 license or if the copyright holder’s requirements are incompatible with the CC BY 4.0 license, please either i) remove the figure or ii) supply a replacement figure that complies with the CC BY 4.0 license. Please check copyright information on all replacement figures and update the figure caption with source information. If applicable, please specify in the figure caption text when a figure is similar but not identical to the original image and is therefore for illustrative purposes only.

Reviewers' comments:

Reviewer's Responses to Questions

**Comments to the Author**

1. Is the manuscript technically sound, and do the data support the conclusions?

Reviewer #1: Partly

Reviewer #2: Yes

2. Has the statistical analysis been performed appropriately and rigorously? 

Reviewer #1: No

Reviewer #2: Yes

3. Have the authors made all data underlying the findings in their manuscript fully available?

Reviewer #1: No

Reviewer #2: Yes

4. Is the manuscript presented in an intelligible fashion and written in standard English?

Reviewer #1: No

Reviewer #2: Yes

5. Review Comments to the Author

Reviewer #1: My overall impression is that the study is of interest as it present novel and important information for the region that surrounds and connects some of the most important world natural heritage areas and this information has the potential to support conservation and management strategies for a range of iconic and threated species and habitats. Although the sampling design has some limitations, they do not preclude the authors from extracting relevant and useful information. However, in my opinion, there are two major issues with the ms as it stands, namely the statistical analysis and the discussion section. The chosen data analysis strategy resulted in a very fragmented analysis which is difficult to follow and makes the overview of the results challenging. I have provided some notes and advise on the ms as believe a modelling approach would have been more adequate. The discussion section is long, fragmented and addresses issues that go beyond the scope of the ms and the data. On the other hand, the conclusions summarize well the results and limitations of the study and is very well written.

For these two main reasons I don’t think this ms in its current form is suitable for publication in PlosOne. Given the relevance and novelty of the study I would recommend a resubmission after major changes.

Reviewer #2: This study used drifting pelagic-BRUVs to characterize the distribution and abundance of large pelagic species (LPS) in an offshore seamount system. The manuscript is well-written and the authors do a great job of demonstrating the utility of these systems in surveying highly mobile LPS in pelagic environments while being candid about the limitations of the study. As the sample sizes/sampling effort are understandably relatively low, I think some of the conclusions come across as a little strong and some parts are a little too speculative. I have made more detailed recommendations below, and would recommend publication of this manuscript following minor revisions.

General comments:

Was a permit required for conducting this field research (and if so, is required in the ‘ethics statement’)?

Given this manuscript serves as a great guide for future studies using these techniques, I think it would also be beneficial to include a few lines in the discussion about how one may design a study to explore if it’s the seamounts themselves resulting in high abundance/diversity of LPS at seamounts, or it being a migratory corridor (e.g. having BRUVs deployed at increasing distances away from the seamount, both in the direction of the ridge and away from the ridge).

Specific comments:

Abstract:

Line 28: add ‘two’ after these so that it reads ‘of these two oceanic marine …’

Introduction:

Line 46: define ‘large’ and ‘pelagic’

Line 54-55: refer to Figure 1 here to help guide the reader

Line 82: The non-expert reader may not understand what is meant by ‘a reduced amount of zeros’. Perhaps replace with ‘a higher probability of recording individuals’, or add in ‘i.e. absences’ after ‘a reduced amount of zeros’.

Line 90: technically scientists are also stakeholders. Replace with ‘non-expert stakeholders’, ‘citizen scientists’ or similar.

Methods:

Line 102: Capitalize ‘south pacific’

Line 127-129: This statement needs to be softened. Many predator species still continue to move into waters deeper than this in this region (as shown by tagging studies).

Line 131: Where was this period on the ENSO scale (i.e. La Nina, El Nino or neutral)?

Line 133-134: It’s slightly confusing to have this statement here, when later on the statistical tests are testing whether abundance and richness differs between shallow and deep seamounts (i.e. why are you testing this if you appear to already know the answer?). Perhaps give a reason here why it is thought that the shallower areas will have increased detection rates (i.e. based on previous studies in other regions), or rephrase.

Line 138: Do you know the average error from the NOAA estimates? This would be useful for having an idea of the error associated with the depths of the deeper seamounts.

Line 158-159: ‘after that time is less significant’ sounds a little funny. Maybe rephrase/explain this a little more.

Line 180-182: Did currents or local subsurface oceanography ever inhibit straight line deployments i.e. did the BRUVs ever drift into each other and/or get entangled together?

Line 193-194: Were these concentrations downloaded for the day/time of deployment at that seamount? What is the spatial resolution/error of these measurements?

Line 228: Specifically state what metric of abundance is being tested here – MaxN/hour?

Results:

Was visibility in the water column ever an issue? Did it ever vary between sites, deployment depth, seamount depth or time of day? Worth stating this here as well as in discussion.

Line 256: is ‘species’ here referring to all species, or just LPS? Be specific.

Line 270: I know they aren’t the focus of this study, but I think it would be better to report these statistics separately for dolphins and sea turtles.

Line 280-282: Given only one other individual of a marine mammal species was observed (one black Pacific sea turtle), I’d consider removing these statistics, or at least making it clear that the remaining 3% of the MaxN was due to one turtle sighting.

Discussion

Line 373: Consider replacing ‘reduced’ with ‘lower’

Line 378: replace ‘were’ with ‘where’

Line 383-385: slightly confusing so please reword. Perhaps replace ‘are needed’ with ‘being needed’.

Line 391: replace ‘one of the’ with ‘some of the’

Line 394-396: switching between using scientific name alone, and common name (scientific name) in this sentence. Use one or the other to remain consistent.

Line 401-403: very cool!

Line 403: Replace ‘even’ with ‘ever’.

Line 417-418: replace ‘despite’ with ‘although’

Line 422: Even between shallow and deeper deployments?

Line 467: and the mobility or speed of each group.

Line 482-483: Again, this is quite a strong statement. Tagging in these regions has not shown limitations of elasmobranchs to these depths here. Consider softening as the casual reader may interpret this as these animals aren’t accessing depths deeper than this.

Line 490: Given the relatively low sample sizes and high standard errors, be cautious here, and perhaps state that greater sample sizes/efforts are needed to further understand/validate these trends. Also, replace ‘can be attributed to’ with ‘may be attributed to’.

Line 500-507: Considering the discussion of the ENSO effect in the method, it would be worth discussing how this may influence species distribution/relative abundance, and discussing the need for annual sampling in order to investigate how this may be driving annual variation.

Line 517-519: Perhaps add the alternative hypothesis here i.e. that the presence of pelagic assemblages here may be due to this area/ridge representing a migratory corridor between Cocos and the Galapagos Islands.

Line 527-531: This sentence is confusing, please reword.

Line 549: explicitly state here which seasons have higher fishing efforts

Tables and Figures:

Table 1: Include the units for the variables measured in the table headings as well as the caption to facilitate easy reading.

Table 2: Replace ‘sripped marlin’ with ‘striped marlin’ and capitialise ‘pacific’ in ‘Black pacific turtle’.

Figure 5: What are the errors bars here? Standard deviation or standard errors?

6. PLOS authors have the option to publish the peer review history of their article (what does this mean?). If published, this will include your full peer review and any attached files.

Reviewer #1: No

Reviewer #2: **Yes: **Samantha Andrzejaczek

---

## [Author Response · Author response to Decision Letter 0]

13 Apr 2021

Reviewer 1 (comment 1): We appreciate the positive comment and agree that the information presented here will be an important reference for future studies towards spatial marine planning in the region

Reviewer 1 (comment 2): we have considered the reviewer’s comment and have added generalized linear models as part of the analysis.

Reviewer 1 (comment 3): we have rewritten most of the discussion section. We consider this version is more fluent and straight to the point.

Reviewer 2 (comment 1): we appreciate the reviewer’s comments

Reviewer 2 (comment 2): we have rewritten the discussion and consider we have addressed this issue

Reviewer 2 (comment 3): We have addressed all recommendations and hope the reviewer find this version for publication

---

## [Decision Letter · Decision Letter 1]

11 May 2021

PONE-D-20-38469R1

A first assessment of the distribution and abundance of large pelagic species at Cocos Ridge seamounts (Eastern Tropical Pacific) using drifting pelagic baited remote cameras

PLOS ONE

Dear Dr. Cambra,

Thank you for submitting your manuscript to PLOS ONE. After careful consideration, we feel that it has merit but does not fully meet PLOS ONE’s publication criteria as it currently stands. Therefore, we invite you to submit a revised version of the manuscript that addresses the points raised during the review process.

Thank you for submitting your revised manuscript PONE-D-20-38469R1 “A first assessment of the distribution and abundance of large pelagic species at Cocos Ridge seamounts (Eastern Tropical Pacific) using drifting pelagic baited remote cameras” to PLOS ONE. One of the initial reviewers could not review the revised version so I found another independent reviewer who returned a similar report highlighted two main issues: 1) a discussion section that is too long and 2) some issues with the statistical analyses. This implies that the revision did not completely addressed the issues highlighted during the initial review process. For these reasons, I do not think your work is currently acceptable for publication. As all reviewers found that the data are interesting and could provide an important contribution for conservation and management in this region, I invite you to thoroughly address the statistical concerns and reduce your discussion in a revised version. However please note that if the same issues are present in the second revision I might have to reject the paper so please try to correctly address these concerns.

Kind regards

Johann

We look forward to receiving your revised manuscript.

Kind regards,Kind regards

Johann

Johann Mourier, Ph.D.

Academic Editor

PLOS ONE

Additional Editor Comments (if provided):

Thank you for submitting your revised manuscript PONE-D-20-38469R1 “A first assessment of the distribution and abundance of large pelagic species at Cocos Ridge seamounts (Eastern Tropical Pacific) using drifting pelagic baited remote cameras” to PLOS ONE. One of the initial reviewers could not review the revised version so I found another independent reviewer who returned a similar report highlighted two main issues: 1) a discussion section that is too long and 2) some issues with the statistical analyses. This implies that the revision did not completely addressed the issues highlighted during the initial review process. For these reasons, I do not think your work is currently acceptable for publication. As all reviewers found that the data are interesting and could provide an important contribution for conservation and management in this region, I invite you to thoroughly address the statistical concerns and reduce your discussion in a revised version. However please note that if the same issues are present in the second revision I might have to reject the paper so please try to correctly address these concerns.

Kind regards

Johann

Reviewers' comments:

Reviewer's Responses to Questions

**Comments to the Author**

1. If the authors have adequately addressed your comments raised in a previous round of review and you feel that this manuscript is now acceptable for publication, you may indicate that here to bypass the “Comments to the Author” section, enter your conflict of interest statement in the “Confidential to Editor” section, and submit your "Accept" recommendation.

Reviewer #2: (No Response)

Reviewer #3: (No Response)

2. Is the manuscript technically sound, and do the data support the conclusions?

Reviewer #2: Yes

Reviewer #3: Partly

3. Has the statistical analysis been performed appropriately and rigorously? 

Reviewer #2: Yes

Reviewer #3: No

4. Have the authors made all data underlying the findings in their manuscript fully available?

Reviewer #2: Yes

Reviewer #3: Yes

5. Is the manuscript presented in an intelligible fashion and written in standard English?

Reviewer #2: Yes

Reviewer #3: Yes

6. Review Comments to the Author

Reviewer #2: Thank you for addressing my comments. I have a few minor comments remaining.

Line 47: All marine species technically inhabit the water column. Maybe elaborate slightly more e.g. upper layers of the water column, and/or give a depth range.

Line 49: As the method for measuring the different species included here varies (e.g. TL versus disc width etc), it may be better to refer to mass here. E.g. Estes et al 2016 referred to marine megafauna as being >45 kg max reported mass, or Andrzejaczek et al 2019 referred to large epipelagic species as >30 kg max reported mass.

Line 107: Capitalize ‘pacific’.

Line 199: Be specific, how many BRUVs units were not included in the analysis?

Line 322: Replace ‘where’ with ‘were’.

Line 398: What was the directionality of the relationship with ‘distance to nearest MPA’? Higher or lower richness and abundance with increasing distance?

Line 510-514: Reword. This currently reads that shallower habitats have less habitat suitability than deeper waters in the ETP, therefore suggesting BRUVS deployments should be deeper to capture LPS.

Line 517-518: Again be specific at the start of the paragraph here about the directionality of this relationship.

Line 540: These results aren’t technically the ‘opposite’. Perhaps state that they are ‘inconsistent with [103]’.

Line 627: The thermocline depth will vary with location and time of year. Perhaps state ‘(here 25 m)’.

Line 637-638: rephrase ‘including a positive association of LPS to seamounts’. This was not explicitly tested here as associations away from seamounts were not also tested here.

Reference list:

Andrzejaczek S, Gleiss AC, Pattiaratchi CB, Meekan MG (2019) Patterns and drivers of vertical movements of the large fishes of the epipelagic. Reviews in Fish Biology and Fisheries 29:335-354

Estes JA, Heithaus MR, McCauley DJ, Rasher DB, Worm B (2016) Megafaunal impacts on structure and function of ocean ecosystems. Annual Review of Environment and Resources 41:83-116

Reviewer #3: Overview

- This study uses drifting pelagic BRUVs to assess large pelagic species along a seamount ridge between the Galapagos Islands and Cocos Islands in the Eastern Tropical Pacific. I appreciate that there is a large undertaking to collecting such data, and the information gathered will be of interest to audiences, with applications for management and conservation.

- In my opinion, there are some issues that are of concern with regards to publication. The main issue is the statistical analysis. The presentation and description of the methods used is confusing, with important details not described making it difficult to follow the processes. The authors use Wilcoxon Rank sum tests to look at individual variable effects on their response variables of either species richness, or relative abundance. Then, depending on the outcome of these tests, some variables were included in a GLM using the same response variables. It is not best practise to multiple hypothesis test, and I don’t understand why relevant variables aren’t included in the model from the outset. If the variable is considered biologically/ecologically/behaviourally relevant, then include it and use the modelling framework to decipher the results. Within the supplementary materials table there is an offset listed as being used, but not described. This is an important feature of the model if used, and as it is stated, is not used correctly as an offset should be logged in this framework. Also, there are two presentations of the models, one showing AIC values highlighting most parsimonious model (Table S2), and a summary table showing p-values (Table 4) neither of which are fully described as to how the final conclusions were made and why two evaluation methods are being used. Two variables were shown to be of importance in the models, but only one is plotted, is there a reason seamount depth was not shown? The other main area is the discussion – I feel it is overly long and quite dis-jointed. There is less contextualising of the main results and more review on the technique as a whole. Due to the reasons outlined, I don’t feel the manuscript is currently at a level suitable for publication.

Detailed points

Lines 28-29: Are you talking about moving between the EEZs or MPAs? Make clearer.

Lines 33-34: Just use teleost rather than size specific.

Lines 125-126: Need a source and reference for the cited temperature.

Lines 189-208: This is quite confusing and hard to follow. Need to consider the ordering and stepwise description of deployments. Quite a lot of averages etc. should these not be in the results.

Lines 192-193: Need to state why you refer to each longline as one deployment. Due to independence of observations?

Lines 199-201: Instead of saying “few” or “less than” state the actual numbers.

Line 201: Should final be total?

Lines 217-221: State the product used, it sounds like MODIS but the details are needed.

Lines 236-239: If feels a shame to be removing data here. Could there be other ways to consider this data? If large numbers are seen, this does not impact being included in analysis of species richness.

Lines 243-244: Can you explain this, how does correlation equal unbiased index?

Lines 257-259: Why do you use negative binomial for one analysis? Should be stated. In table S2 there is an offset used, which is not stated in the methods – this is a very important piece of information to include. How was this incorporated? In the table it is not stated as being a log of the soak time. IS this correct? The offset should be logged in these model algorithms.

Table 2: This table is confusing – The authors described that certain variables were not included in the modelling approach, however they appear in the table of model variables, and include a symbol which states they were not included. The table should only include the variables tested.

Lines 289-292: This is the first mention of SST, what product was used for this? How were VIF, positive redundancy, and multi-collinearity calculated? What variables was SST correlated with, and why were they chosen over SST?

Lines 293-295: Are the land points single coordinates? Where are these located, centre of the islands? Could a land shapefile of the islands not be used to get a more accurate measure of distance from land?

Lines 297-298: Differences in AIC were stated as being used, but not fully explained. How were AIC scores used to determine best model? Was there a threshold?

Lines 308: Should 132 be 150?

Lines 315-317: What species, what IUCN classification?

Lines 318-323: What is the metric ind per hour? This hasn’t been explained. Is this the same as MaxN per hour?

Lines 394-405: Direction of the relationships need to be described. Variables are being highlighted as having an effect, however the reader doesn’t know which way e.g. distance to nearest MPA is important, but are there more fish nearer or further away?

Lines 458-460: Is this really the first video evidence of this behaviour? Or at these seamounts specifically?

Lines 544-545: Is drift speed actually included in any model? Track distance is included but not speed from what I can see.

Fig. 1: EZZ should be EEZ.

7. PLOS authors have the option to publish the peer review history of their article (what does this mean?). If published, this will include your full peer review and any attached files.

Reviewer #2: **Yes: **Samantha Andrzejaczek

Reviewer #3: No

---

## [Author Response · Author response to Decision Letter 1]

29 Jun 2021

Reply to review comment from reviewer 1

COMMENT 1: Thank you for addressing my comments. I have a few minor comments remaining.

ANSWER 1: You are welcome. Thanks for your revision. We have addressed all minor comments and hope it meets the reviewer's requirements.

Reply to review comments from reviewer 2

COMMENT 1: In my opinion, there are some issues that are of concern with regards to publication. The main issue is the statistical analysis. The presentation and description of the methods used is confusing, with important details not described making it difficult to follow the processes

ANSWER 1: We have described all the statistical methods with more detail and with the appropriate references. By deleting the Wilcoxon Rank sum tests prior to perform the models we believe the presented information is now clearer. 

COMMENT 2: The authors use Wilcoxon Rank sum tests to look at individual variable effects on their response variables of either species richness, or relative abundance. Then, depending on the outcome of these tests, some variables were included in a GLM using the same response variables. It is not best practice to multiple hypothesis test, and I don’t understand why relevant variables aren’t included in the model from the outset. If the variable is considered biologically/ecologically/behaviourally relevant, then include it and use the modelling framework to decipher the results.

ANSWER 2: Initially, we didn’t want to saturate the models with too many categorical variables since our data set is small. However, we have followed the reviewer’s advice of not using non-parametric tests to decide which variables to add in the models. Instead, we added all variables in the models to show their significance. We have also transformed two predictor variables that didn’t have a normal distribution and have add the log of soak time, instead of only soak time as an offset. These changes have resulted into better fitted models presented in this version.

COMMENT 3: Within the supplementary materials table there is an offset listed as being used, but not described. This is an important feature of the model if used, and as it is stated, is not used correctly as an offset should be logged in this framework

ANSWER 3: We have used the log of soak time following the reviewer’s comment.

COMMENT 4: Also, there are two presentations of the models, one showing AIC values highlighting most parsimonious model (Table S2), and a summary table showing p-values (Table 4) neither of which are fully described as to how the final conclusions were made and why two evaluation methods are being used.

ANSWER 4:We have better specified how we used the AIC to select the optimal model in the methods. Additionally, we have added this sentence in the results section based on this comment: “Summary results from selected models with the level of significance of each predictor are shown in Table 4. The model selection process based on maximum likelihood ratio tests and AIC for Poisson and Negative binomial models is shown in S2 Table”. 

COMMENT 5: Two variables were shown to be of importance in the models, but only one is plotted, is there a reason seamount depth was not shown?

ANSWER 5: Since the best fitted Negative binomial model for LPS abundance had two significant predictors we have combined them in the same graph. We believe this is a better way to show the effect of both predictors together instead of presenting two different graphs. 

COMMENT 6: The other main area is the discussion – I feel it is overly long and quite dis-jointed. There is less contextualizing of the main results and more review on the technique as a whole.

ANSWER 6: We have reduced the discussion quite a bit (approximately 1000 words less) and reorganized the information to be more organized.

---

## [Decision Letter · Decision Letter 2]

24 Aug 2021

PONE-D-20-38469R2

A first assessment of the distribution and abundance of large pelagic species at Cocos Ridge seamounts (Eastern Tropical Pacific) using drifting pelagic baited remote cameras

PLOS ONE

Dear Dr. Cambra,

Thank you for submitting your manuscript to PLOS ONE. After careful consideration, we feel that it has merit but does not fully meet PLOS ONE’s publication criteria as it currently stands. Therefore, we invite you to submit a revised version of the manuscript that addresses the points raised during the review process.

Thank you for submitting your revised manuscript PONE-D-20-38469R2 “A first assessment of the distribution and abundance of large pelagic species at Cocos Ridge seamounts (Eastern Tropical Pacific) using drifting pelagic baited remote cameras” to PLOS ONE. One reviewer still highlighted some issues with the statistical analyses while the other was satisfied with the revision. Before taking a decision I therefore invited another reviewer who also found some issues in the modelling approach that can be fixed relatively easily. The main issue belongs to the fact that MaxN hr-1, the response variable is a rate so poisson and negative binomial error structures are no longer appropriate. Based on reviewers’ advices, you may have two main option to tackle this:

1) As you have already accounted for soak time in converting MaxN to MaxN hr-1, you could remove the offset of soak time and fit the model with a more appropriate distribution.

2) you could also use the offset of soak time but use the raw MaxN value rather than MaxN hr-1. The raw MaxN value is a true count (integer/discrete) and more appropriate for a Poisson distribution. 

based on reviewers’ suggestions I think that one way to determine relative support for one model can be to calculate the differences between its AICc and the smallest AICc (ΔAICc) and scaling these differences into model weights (wAICc). All models with values of ΔAICc ≤ 2 can be presented, since values within this threshold can have similar explanatory power. Another step after this is that those models can actually be averaged together to get a final output. All of these techniques can be implemented in the MuMIn package for example.

If you can revise your models following these recommendations and respond to the comments of the two reviewers, I think the statistical approach would be more robust and the paper of greater quality.

Kind regards

Johann

We look forward to receiving your revised manuscript.

Kind regards,

Johann Mourier, Ph.D.

Academic Editor

PLOS ONE

Additional Editor Comments (if provided):

Thank you for submitting your revised manuscript PONE-D-20-38469R2 “A first assessment of the distribution and abundance of large pelagic species at Cocos Ridge seamounts (Eastern Tropical Pacific) using drifting pelagic baited remote cameras” to PLOS ONE. One reviewer still highlighted some issues with the statistical analyses while the other was satisfied with the revision. Before taking a decision I therefore invited another reviewer who also found some issues in the modelling approach that can be fixed relatively easily. The main issue belongs to the fact that MaxN hr-1, the response variable is a rate so poisson and negative binomial error structures are no longer appropriate. Based on reviewers’ advices, you may have two main option to tackle this:

1) As you have already accounted for soak time in converting MaxN to MaxN hr-1, you could remove the offset of soak time and fit the model with a more appropriate distribution.

2) you could also use the offset of soak time but use the raw MaxN value rather than MaxN hr-1. The raw MaxN value is a true count (integer/discrete) and more appropriate for a Poisson distribution.

based on reviewers’ suggestions I think that one way to determine relative support for one model can be to calculate the differences between its AICc and the smallest AICc (ΔAICc) and scaling these differences into model weights (wAICc). All models with values of ΔAICc ≤ 2 can be presented, since values within this threshold can have similar explanatory power. Another step after this is that those models can actually be averaged together to get a final output. All of these techniques can be implemented in the MuMIn package for example.

If you can revise your models following these recommendations and respond to the comments of the two reviewers, I think the statistical approach would be more robust and the paper of greater quality.

Kind regards

Johann

Reviewers' comments:

Reviewer's Responses to Questions

**Comments to the Author**

1. If the authors have adequately addressed your comments raised in a previous round of review and you feel that this manuscript is now acceptable for publication, you may indicate that here to bypass the “Comments to the Author” section, enter your conflict of interest statement in the “Confidential to Editor” section, and submit your "Accept" recommendation.

Reviewer #2: All comments have been addressed

Reviewer #3: (No Response)

Reviewer #4: (No Response)

2. Is the manuscript technically sound, and do the data support the conclusions?

Reviewer #2: Yes

Reviewer #3: Partly

Reviewer #4: Yes

3. Has the statistical analysis been performed appropriately and rigorously? 

Reviewer #2: Yes

Reviewer #3: No

Reviewer #4: Yes

4. Have the authors made all data underlying the findings in their manuscript fully available?

Reviewer #2: (No Response)

Reviewer #3: Yes

Reviewer #4: Yes

5. Is the manuscript presented in an intelligible fashion and written in standard English?

Reviewer #2: Yes

Reviewer #3: Yes

Reviewer #4: Yes

6. Review Comments to the Author

Reviewer #2: (No Response)

Reviewer #3: PONE-D-20-38469R2 – A first assessment of the distribution and abundance of large pelagic species at Cocos Ridge seamounts (Eastern Tropical Pacific) using drifting pelagic baited remote cameras.

Overview

- This is the second time I have reviewed this, and the authors have made good efforts to improve the writing of the manuscript, especially the length and focus of the discussion. Unfortunately I don’t feel the authors have addressed my original concerns with regards to the statistical analysis to warrant publication.

- My comments are focussed on the statistical analysis as it is my main concern with the manuscript, and is still very confused and often applying techniques and methods inappropriately.

- A response variable of relative abundance is tested using MaxN hr-1. The authors have created a rate as the response variable and so poisson and negative binomial error structures are no longer appropriate. The best approach would be to model the rate using an offset, however this needs to be used in conjunction with a count response variable – in this case MaxN. The authors have included an offset of logged soak time as an offset, based on comments from the previous version, but with a response variable that is already a rate, therefore not applying this technique correctly, and suggesting not fully understanding the reasoning behind the approach.

- Fixed effects should not be transformed.

- The authors use the z-statistic from the model summary tables as a measure of significance to remove variables from their model. The values given in the summary tables are not corrected and therefore should not be trusted as a test of significance. This is also a pseudo stepwise model selection, which is a highly criticised approach.

- The authors then use AIC (should it be AICc due to small sample sizes?), which is a quite different approach and the two are not complementary. This also applies to comparing AIC ranked models with an ANOVA to obtain a p-value to select a model, there are methods more appropriate to AIC ranked model sets.

- The authors state they removed the use of Wilcoxon Rank sum tests to look at effects on relative abundance, however there is still a statement of their use for categorical variables (lines 289-292), even after these are included in the GLMs. These variables are being used to test their influence on relative abundance in a model, therefore this creates a multiple comparison problem, and I’m not sure what is to be gained from getting another measurement of effect of these variables.

Reviewer #4: This paper uses drifting pelagic BRUVS to provide the first assessment of the distribution and abundance of large pelagic species at Cocos Ridge seamounts in the Eastern Tropical Pacific. Overall, I feel the paper is well written and the authors do a good job in the discussion of balancing out their conclusions/interpretations and acknowledging the limitations of their data set. The use of drifting pelagic BRUVS is novel and it is nice to see these protocols being extended to new areas of interest.

I reviewed a revision of this manuscript and feel that the authors did a good job of addressing the concerns of the two initial reviewers. Specifically, I feel that the change to include all relevant variables in the GLMs from the outset has improved the analyses. I also feel that the authors adequately addressed the concerns regarding the addition of more detail surrounding the methods. With that said, I do still have some minor concerns with the analysis component, but I do not feel these would take that much effort to fix as it more pertains to issues around the interpretation of the 'top model'. As such, my recommendation is that this manuscript would be suitable for publication in PLOS ONE after some minor revisions. If the authors wish to discuss any of my comments or would like further clarification then I am happy for them to contact me. Justin Rizzari (Justin.Rizzari@deakin.edu.au).

See below for comments:

1) I did not see Figure 4 in any of the documents. In the manuscript it goes from Figure 3 (the pictures of individual species) to Figure 5 (depth and MPA comparisons). I also did not see Figure S4 in any of the documents. The Figure versions I received were also quite blurry, although I suspect this may be an issue with uploading it to the system

2) The authors state the best fitting GLM for LPS richness was one that included only seamount depth. Upon looking at the table though it would seem that based on AIC values that model 1 was actually the best. I realise that model 2 has a higher residual deviance and that the authors stated that in the case of models with similar AIC values of one another that the most parsimonious model would be deemed the best fit. However, I would just suggest pointing this out in the results section to reiterate it. Another alternative would be to do model averaging, which is a commonly employed approach for models within 2 AIC values of one another. In the case of this paper it would be the top 3 for LPS richness and top 2 for MaxN. Furthermore, given that model 1 for richness also had MPA distance as a factor I think it would be better to make Figure 5 show depth and MPA distance results for species richness similar to what has been done for MaxN.

3) Lines 486-494: The authors state that environmental drivers were not significant predictors of LPS richness or abundance. However, chl a was actually important in model 2 for MaxN, which was within 2 AIC values of model 1. At the very least some discussion to this would seem appropriate.

4) Lines 495 -496. The authors also state that camera depth of time of day had no significant effect on LPS richness or abundance. However, for LPS richness 'diel' was actually an important factor in model 3, which was within 2 AIC values. Similar to point #3 above I feel that some discussion of this is warranted.

7. PLOS authors have the option to publish the peer review history of their article (what does this mean?). If published, this will include your full peer review and any attached files.

Reviewer #2: No

Reviewer #3: No

Reviewer #4: **Yes: **Justin Rizzari

---

## [Author Response · Author response to Decision Letter 2]

1 Oct 2021

EDITOR COMMENTS

COMMENT 1: The main issue belongs to the fact that MaxN hr-1, the response variable is a rate so poisson and negative binomial error structures are no longer appropriate. Based on reviewers’ advices, you may have two main option to tackle this:

1) As you have already accounted for soak time in converting MaxN to MaxN hr-1, you could remove the offset of soak time and fit the model with a more appropriate distribution

2) You could also use the offset of soak time but use the raw MaxN value rather than MaxN hr-1. The raw MaxN value is a true count (integer/discrete) and more appropriate for a Poisson distribution

ANSWER 1: We have already clarified in the comment 2 of the reviewer 3 that the model was actually performed using MaxN as a response variable and not MaxN hr-1. However, we made the mistake of writing MaxN hr-1 in the text of the manuscript instead of MaxN when describing the models. We have already changed MaxN hr-1 by MaxN or simply added MaxN to clarify in the lines 249, 394 and 431.

COMMENT 2: One way to determine relative support for one model can be to calculate the differences between its AICc and the smallest AICc (ΔAICc) and scaling these differences into model weights (wAICc). All models with values of ΔAICc ≤ 2 can be presented, since values within this threshold can have similar explanatory power. Another step after this is that those models can actually be averaged together to get a final output. All of these techniques can be implemented in the MuMIn package for example

ANSWER 2: We have incorporated model averaging following the editor’s comment (see answer to question 4 of reviewer 3).

REVIEWER 3 COMMENTS

COMMENT 1: This is the second time I have reviewed this, and the authors have made good efforts to improve the writing of the manuscript, especially the length and focus of the discussion. Unfortunately I don’t feel the authors have addressed my original concerns with regards to the statistical analysis to warrant publication. My comments are focused on the statistical analysis as it is my main concern with the manuscript, and is still very confused and often applying techniques and methods inappropriately.

ANSWER 1: We have incorporate all suggestions made by the reviewer regarding the statistical analysis and expect to have addressed his/her concerns. 

COMMENT 2: A response variable of relative abundance is tested using MaxN hr-1. The authors have created a rate as the response variable and so poisson and negative binomial error structures are no longer appropriate. The best approach would be to model the rate using an offset, however this needs to be used in conjunction with a count response variable – in this case MaxN. The authors have included an offset of logged soak time as an offset, based on comments from the previous version, but with a response variable that is already a rate, therefore not applying this technique correctly, and suggesting not fully understanding the reasoning behind the approach.

ANSWER 2: We understand the reviewer’s concern and would like to clarify that the model was performed using MaxN as a response variable and not MaxN hr-1. Our error was writing MaxN hr-1 in the text of the manuscript instead of MaxN when describing the models. In fact, in the S2 Table we present all performed models using MaxN as a response variable. We have already changed MaxN hr-1 by MaxN or simply add MaxN to clarify in the lines 249, 394 and 431 of the manuscript.

COMMENT 3: Fixed effects should not be transformed

ANSWER 3: New models were performed without transforming predictor variables based on the reviewer’s comment and the information in the manuscript describing the transformation of predictor variables was deleted (lines 278 – 282). 

COMMENT 4: The authors use the z-statistic from the model summary tables as a measure of significance to remove variables from their model. The values given in the summary tables are not corrected and therefore should not be trusted as a test of significance. This is also a pseudo stepwise model selection, which is a highly criticized approach. The authors then use AIC (should it be AICc due to small sample sizes?), which is a quite different approach and the two are not complementary. This also applies to comparing AIC ranked models with an ANOVA to obtain a p-value to select a model, there are methods more appropriate to AIC ranked model sets.

ANSWER 4: Model averaging was used instead of the null hypothesis significance testing to compare models and measure the probability of each model to be the best model. All the information regarding the use of the z-statistic and all information from the summary tables has been deleted. Instead we added this information in the methods: “Model selection was based on the small sample-corrected Akaike’s information criterion (AICc). This approach has been suggested as a useful option for small samples where the ratio of observations to model parameters is low (e.g N/K < 40) [78,79]. Models were ranked based on minimum AICc, detailing changes in AICc with respect to the top ranked model (∆AICc) and model weights (wAICc) [80,81]. Models with values of ΔAICc ≤ 2 were presented, since values within this threshold can have similar explanatory power [81,82]. Model weights were computed as a measure of each model’s strength of evidence where the smaller the wAICc, the lower probability the model is true [79]. The cumulative wAICc was used to identify a 95% confidence set of models and to measure the relative importance of each variable [65]. The larger the sum of the weight value (∑wi), the more important the variable is relative to the other variables [65,83]. Residual deviance and GLM diagnostic plots of standard residuals were used to evaluate the goodness of fit of the resulting models and to determine wheatear models assumptions were met [65]”. In the results section we have changed Table 3 from showing the summary results from the best models to comparing the models with values of ΔAICc ≤ 2. We have discussed each of these models based on their wAICc, the relative importance of its predictor variables and its residual deviance.

COMMENT 5: The authors state they removed the use of Wilcoxon Rank sum tests to look at effects on relative abundance, however there is still a statement of their use for categorical variables (lines 289-292), even after these are included in the GLMs. These variables are being used to test their influence on relative abundance in a model, therefore this creates a multiple comparison problem, and I’m not sure what is to be gained from getting another measurement of effect of these variables

ANSWER 5: The Wilcoxon Rank sum test was originally used to compare differences in relative abundance of each ecological group between the categorical predictor variables (seamount depth levels, camera depth levels and diel) and not to test their influence on relative abundance. However, we acknowledge that the way the information was presented it was confusing and that these results are not stated as an objective of this paper. Therefore, we have deleted this analysis and the associated figure.

REVIEWER 4 COMMENTS

COMMENT 1: I did not see Figure 4 in any of the documents. In the manuscript it goes from Figure 3 (the pictures of individual species) to Figure 5 (depth and MPA comparisons). I also did not see Figure S4 in any of the documents. The Figure versions I received were also quite blurry, although I suspect this may be an issue with uploading it to the system

ANSWER 1: We agree there were some errors regarding the Figures citation in the manuscript. We have made the following changes to fix the errors: 1) We have included the caption for Figure 1 (line 115) since that was missing; 2) We have changed Fig 1 for Fig 4 in line 377 since the figure we present is the number 4; 3) We have changed S4 Fig for S3 Fig in line 410. Additionally we have added new supplementary figures (S4 and S5 figures) to show results from the new models (see next comment). 

COMMENT 2: The authors state the best fitting GLM for LPS richness was one that included only seamount depth. Upon looking at the table though it would seem that, based on AIC values, model 1 was actually the best. I realize that model 2 has a higher residual deviance and that the authors stated that in the case of models with similar AIC values of one another that the most parsimonious model would be deemed the best fit. However, I would just suggest pointing this out in the results section to reiterate it. Another alternative would be to do model averaging, which is a commonly employed approach for models within 2 AIC values of one another. In the case of this paper it would be the top 3 for LPS richness and top 2 for MaxN. Furthermore, given that model 1 for richness also had MPA distance as a factor I think it would be better to make Figure 5 show depth and MPA distance results for species richness similar to what has been done for MaxN.

ANSWER 2: We have used the alternative suggested by the author and instead of selecting one single best model we used model averaging to compare models and to measure the probability of each model to be the best model. In this new version we have discussed each of the models with values of ΔAICc ≤ 2 based on their wAICc, the relative importance of its predictor variables and its residual deviance. See the answer to comment 4 of the previous reviewer to see how the information has been presented in the new version of the manuscript. 

Seamount depth was the only significant variable on LPS richness. However the first ranked model for LPS richness included seamount depth and minimum distance to nearest MPA as predictor variables. Therefore, and following the reviewer’s comment, we have included both variables in figure 5; one graph only showing the effect of seamount depth and a second graph in the same figure showing the effect of both variables.

COMMENT 3: Lines 486-494: The authors state that environmental drivers were not significant predictors of LPS richness or abundance. However, chl a was actually important in model 2 for MaxN, which was within 2 AIC values of model 1. At the very least some discussion to this would seem appropriate

ANSWER 3: We have made some modifications regarding the discussion of environmental parameters based on new results from model averaging (lines 525 – 549) where we recognize the potential influence of chl-a on LPS abundance since it was included in the third ranked candidate model with Δi values ≤ 2 (line 544).

COMMENT 4: Lines 495 -496. The authors also state that camera depth of time of day had no significant effect on LPS richness or abundance. However, for LPS richness ‘diel’ was actually an important factor in model 3, which was within 2 AIC values. Similar to point #3 above I feel that some discussion of this is warranted

ANSWER 4: Similar to what we answered in the previous comment, we have also recognized the potential importance of time of day and camera depth level for LPS richness and abundance since they were included in models with Δi values ≤ 2. We also present and discuss the higher relative importance of time of day compared to other non-significant predictor variables (lines 525 – 549).

---

## [Decision Letter · Decision Letter 3]

12 Oct 2021

A first assessment of the distribution and abundance of large pelagic species at Cocos Ridge seamounts (Eastern Tropical Pacific) using drifting pelagic baited remote cameras

PONE-D-20-38469R3

Dear Dr. Cambra,

We’re pleased to inform you that your manuscript has been judged scientifically suitable for publication and will be formally accepted for publication once it meets all outstanding technical requirements.

Kind regards,

Johann Mourier, Ph.D.

Academic Editor

PLOS ONE

Additional Editor Comments (optional):

Reviewers' comments:

Reviewer's Responses to Questions

**Comments to the Author**

1. If the authors have adequately addressed your comments raised in a previous round of review and you feel that this manuscript is now acceptable for publication, you may indicate that here to bypass the “Comments to the Author” section, enter your conflict of interest statement in the “Confidential to Editor” section, and submit your "Accept" recommendation.

Reviewer #3: All comments have been addressed

Reviewer #4: All comments have been addressed

2. Is the manuscript technically sound, and do the data support the conclusions?

Reviewer #3: Yes

Reviewer #4: Yes

3. Has the statistical analysis been performed appropriately and rigorously? 

Reviewer #3: Yes

Reviewer #4: Yes

4. Have the authors made all data underlying the findings in their manuscript fully available?

Reviewer #3: Yes

Reviewer #4: Yes

5. Is the manuscript presented in an intelligible fashion and written in standard English?

Reviewer #3: Yes

Reviewer #4: Yes

6. Review Comments to the Author

Reviewer #3: The authors have done a good job re-analysing, wording, and presenting their methods and results.

I would caution against the use of the word "significant" when describing results from this study when ranking of a model set by AICc is presented with no test of significance.

Reviewer #4: (No Response)

7. PLOS authors have the option to publish the peer review history of their article (what does this mean?). If published, this will include your full peer review and any attached files.

Reviewer #3: No

Reviewer #4: **Yes: **Justin Rizzari

---

## [Editor Report · Acceptance letter]

10 Nov 2021

PONE-D-20-38469R3 

A first assessment of the distribution and abundance of large pelagic species at Cocos Ridge seamounts (Eastern Tropical Pacific) using drifting pelagic baited remote cameras 

Dear Dr. Cambra:

I'm pleased to inform you that your manuscript has been deemed suitable for publication in PLOS ONE. Congratulations! Your manuscript is now with our production department. 

Kind regards, 

on behalf of

Dr. Johann Mourier 

Academic Editor

PLOS ONE